# The Ins and Outs of Clusterin: Its Role in Cancer, Eye Diseases and Wound Healing

**DOI:** 10.3390/ijms241713182

**Published:** 2023-08-24

**Authors:** Christelle Gross, Louis-Philippe Guérin, Bianca G. Socol, Lucie Germain, Sylvain L. Guérin

**Affiliations:** 1Centre de Recherche en Organogénèse Expérimentale de l’Université Laval/LOEX, Québec City, QC G1V 0A6, Canada; christelle.gross.1@ulaval.ca (C.G.); bianca.socol.1@ulaval.ca (B.G.S.); lucie.germain@fmed.ulaval.ca (L.G.); 2Centre de Recherche du CHU de Québec, Axe Médecine Régénératrice, Québec City, QC G1J 1Z4, Canada; 3Département d’Ophtalmologie, Faculté de Médecine, Université Laval, Québec City, QC G1V 0A6, Canada; 4Service d’Ophtalmologie, Université de Sherbrooke, Sherbrooke, QC J1K 2R1, Canada; louis-philippe.guerin@usherbrooke.ca; 5Département de Chirurgie, Faculté de Médecine, Université Laval, Québec City, QC G1V 0A6, Canada

**Keywords:** clusterin, gene expression, cancer, eye, cornea, wound healing

## Abstract

Clusterin (CLU) is a glycoprotein originally discovered in 1983 in ram testis fluid. Rapidly observed in other tissues, it was initially given various names based on its function in different tissues. In 1992, it was finally named CLU by consensus. Nearly omnipresent in human tissues, CLU is strongly expressed at fluid–tissue interfaces, including in the eye and in particular the cornea. Recent research has identified different forms of CLU, with the most prominent being a 75–80 kDa heterodimeric protein that is secreted. Another truncated version of CLU (55 kDa) is localized to the nucleus and exerts pro-apoptotic activities. CLU has been reported to be involved in various physiological processes such as sperm maturation, lipid transportation, complement inhibition and chaperone activity. CLU was also reported to exert important functions in tissue remodeling, cell–cell adhesion, cell–substratum interaction, cytoprotection, apoptotic cell death, cell proliferation and migration. Hence, this protein is sparking interest in tissue wound healing. Moreover, *CLU* gene expression is finely regulated by cytokines, growth factors and stress-inducing agents, leading to abnormally elevated levels of CLU in many states of cellular disturbance, including cancer and neurodegenerative conditions. In the eye, CLU expression has been reported as being severely increased in several pathologies, such as age-related macular degeneration and Fuch’s corneal dystrophy, while it is depleted in others, such as pathologic keratinization. Nevertheless, the precise role of CLU in the development of ocular pathologies has yet to be deciphered. The question of whether CLU expression is influenced by these disorders or contributes to them remains open. In this article, we review the actual knowledge about CLU at both the protein and gene expression level in wound healing, and explore the possibility that CLU is a key factor in cancer and eye diseases. Understanding the expression and regulation of CLU could lead to the development of novel therapeutics for promoting wound healing.

## 1. Introduction

### 1.1. Background

CLU was first described in 1983 by Blaschuk et al. [1]. Initially identified as a secreted glycoprotein in ram rete testis fluid that enhanced aggregation of cells in vitro [1], it was later detected in other tissues under different names according to the function investigated: sulphated glycoprotein-2 (SGP-2) (main secreted product of rat Sertoli cells) [2], testosterone-repressed prostatic messenger 2 (TRPM-2) (rat prostatic involution following chemical or surgical castration) [3], T64 (quail mRNA induced in neuroretinal cells by Rous sarcoma virus) [4] and gp-III (bovine constituent of chromaffin granules) [5].

The human homolog of clusterin has been given numerous acronyms, including SP-40,40 protein (identified in the human SC5b complex of complement and in immune deposits in glomerulonephritis) [6], apolipoprotein J [7] and complement lysis inhibitor (CLI) [8] (human lipoprotein associated with high- and very-high-density lipoproteins in human serum and plasma), pADHC-9 (human mRNA from the hippocampus of Alzheimer’s disease patients) [9] and pTB16 (human mRNA highly expressed in epileptic foci) [10].

Ultimately, a consensus was reached at the inaugural international workshop held in 1992 in Cambridge, England. All participants agreed on the common name of clusterin (HUGO nomenclature: CLU).

### 1.2. CLU Expression

The human *CLU* gene has at least 201 orthologs in various species, 17 transcript variants and encodes for CLU protein [11], a nearly ubiquitously expressed and highly conserved protein [12,13,14].

*CLU* gene expression was found to increase from gestation to adulthood in humans [12]. In mice, histological examination of embryos 14.5, 16.5 and 18.5 days old revealed that *CLU* mRNA is widely expressed during murine embryogenesis. *CLU* mRNA is present in cells derived from all three germ layers and most abundant in the epithelial cells of developing organs. Early expressed, from 12.5 days of development onwards, the *CLU* mRNA is widely expressed in epithelia of the developing auditory, olfactory and visual apparatus [15,16]. However, *CLU* gene expression is not restricted to developing epithelia, as all epithelia do not express the *CLU* gene. Conversely, non-epithelial cell types also express *CLU* (such as cartilage, bone, muscle, etc.) [15,16].

### 1.3. Functions

As a result of its wide protein distribution, CLU has been shown to be present and/or involved in numerous physiopathological processes, including sperm maturation [2], complement inhibition [8] and lipid transportation [14]. Additionally, other functions involving secretory pathways have been attributed to CLU, such as tissue remodelling and membrane recycling [17,18]. Cell aggregation and adhesion are also mediated by CLU as it plays a role in cell–cell and cell–substratum interactions [19,20]. Furthermore, CLU has been implicated in apoptosis [21,22,23,24,25,26]. Whether it is as a pro- or anti-apoptotic molecule is still a matter of considerable ambiguity. More recent data have provided evidence that CLU has a chaperone-like activity similar to that of small heat shock proteins (sHSPs), as it can bind extracellular unrelated molecules [27,28,29]. Sequence analysis revealed that CLU contains three long regions with putative amphipathic *α*-helices providing the ability to bind to a variety of molecules (membrane receptors, bacterial proteins, heparin, other apolipoproteins, IgGs and amyloid-forming molecules, as well as partially folded stressed proteins) [30]. Studies have revealed that sCLU functions as an extracellular chaperone that binds to hydrophobic regions of partially unfolded proteins and inhibits, via an ATP-independent mechanism, their stress-induced precipitation [27,28,31,32]. On the other hand, CLU also functions intracellularly, as shown by its cytoplasmic distribution and its interaction with signal transduction and DNA repair proteins [22,33,34,35,36,37]. Furthermore, CLU is also found in the nucleus (named nCLU) following irradiation and acts as a pro-apoptotic protein by interacting with other proteins involed in apoptosis [23,38].

## 2. Gene and Protein Structure

### 2.1. CLU Gene

Human *CLU* is a single-copy gene located on the 8p21.1 region of chromosome 8. The *CLU* gene is organized in nine exons of variable sizes (from 126 to 412 bp), spanning a region over 17,877 bp and coding a transcript of approximatively 2kb [11,12,39], and this gene structure is also well conserved across species (mouse and rat) [18,40,41]. Although many studies have been conducted to better understand the biosynthesis and regulation of CLU in the last few decades, only basic understanding about *CLU*’s gene expression and regulation has been reported. Before 2006, it was believed that the *CLU* gene coded for a unique transcript of 1,9kb containing two in-frame ATG start sites (the first being located in exon 2) (human clusterin coding sequence, GenBank accession no. M64722). However, more recent reports suggest that many more transcripts can code for CLU. At least 17 different mRNA variants (coding or not for CLU proteins) have been reported, of which three major transcripts are more frequently expressed in human cells (NCBI database: isoform 1 NM_001831.4, isoform 2 NR_038335.2 and isoform 3 NR_045494.1) [11,42]. The existence of different transcripts probably originates from alternative starting sites. Indeed, all transcripts contain nine exons and eight introns; exon 1 is unique to each transcript, indicating distinct transcription start sites and ATG (Figure 1—exon 1a and 1c), whereas exons 2–9 are common to all isoforms. All isoforms yield the same open reading frame, with the starting ATG being located in exon 2, just in front of the endoplasmic reticulum (ER)-signal peptide sequence [43,44,45]. The resulting proteins were predicted to be secreted. All variants have a third ATG located in exon 3 downstream of the ER signal peptide sequence. Using 5′ RACE PCR, researchers demonstrated that the 5′-end of isoform 1 differs from NM_001831 as it contains a 5′-extended exon 1 sequence [39,43]. More recently, an alternative messenger was found by Leskov et al. [23]. The authors identified a new variant of *CLU* isoform 1 lacking exon 2 (isoform 1 [Δex2]), and in which both exon 1 and exon 3 are directly joined together. It has been proposed that stress conditions can favor the transcription and relocation in the nucleus of *CLU* truncated mRNA lacking exon 2 [22,23,46]. This transcript, lacking the ATG start site from exon 2, is translated from the third ATG located in exon 3, resulting in the production of a shorter CLU protein. As this protein is depleted of the ER signal peptide, it is predicted to be localized in the nucleus [43,44,45,47].

### 2.2. CLU Protein and Biosynthesis

Like other eukaryotic genes, *CLU* can produce different mature protein isoforms; two types of proteins have been identified in mammals, a first glycosylated-secretory form, sCLU, and a second non-glycosylated and non-secretory (or nuclear) form, produced by alternative splicing [22,47,48]. However, which transcript is related to which protein is always elusive. Furthermore, contradictory functions of CLU have been observed: cytoprotective versus cell death [23,28,49,50]. Translation of mRNA isoform 2 begins at the ATG in the common exon 2 immediately upstream of the ER signal peptide (+1 to +67). This transcript results in the translation of a 449 amino-acids-long precursor protein (Figure 2). The start codon of isoforms 1 and 3 is located in the alternative first exon, which is in frame with the ATG from exon 2. Thus, the precursor sequence for the isoforms 1 and 3 is identical to that of isoform 2, except that the protein encoded by isoforms 1 and 3 contains 52 and 11 additional amino acids at its amino terminus, respectively [39] (Figure 1).

The full-length CLU precursor protein translated from the transcript is 449 amino acids long with 10 Cys residues, 6 potential glycosylation sites and a leader sequence [51,52]. After post-translational modification, this precursor is cleaved into the mature protein linked in antiparallel orientation by five disulfide bridges near their centers, forming a molecule in which the core is flanked by three amphipathic *α*-helices (amino acids 172 to 189, 243 to 259 and 423 to 440) and two coiled-coil *α*-helices (amino acids 42 to 98 and 319 to 349) (Figure 2, in dark blue and green, respectively) [14,23,30].

### 2.3. CLU Protein Isoforms

#### 2.3.1. CLU Secretory Isoform (sCLU)

In the early stages of maturation, CLU mRNA is translated in the cytoplasm into an unfolded 50 kDa precursor protein (psCLU) [23]. Then, this precursor, which contains an N-terminal 22-amino-acid signal peptide, is directed to the endoplasmic reticulum by the leader signaling sequence [23,43]. psCLU removal of the ER signal peptide occurs before it is folded and glycosylated at six glycosylation sites (Asn86, 103, 145, 291, 354 and 374) resulting in a 60 kDa sized precursor protein [51,53]. The glycosylated precursor is then processed by the Golgi apparatus for heavy glycosylation and proteolytic cleavage into α and β subunits at an internal site located between Arg227 and Ser228 [48,54]. The α and β chains are then reassembled by five disulfide bonds in an anti-parallel manner (αCys102-βCys313, αCys113-βCys305, αCys116-βCys302, αCys121-βCys295 and αCys129-βCys285), leading to a mature heterodimeric protein of 75–80 kDa (constituted of two 40 kDa subunits), which is then destined for secretion (sCLU) (Figure 2 and Figure 3) [52,55]. Secretion occurs from secretory vesicles (as previously described) or through non-regulated routes (unconventional protein secretion pathways where proteins bypass intermediate compartments involved in secretion or exocytosis, such as the Golgi apparatus) [54]. Uptake and degradation may be mediated by the endocytic receptor gp330/megalin (LRP-2), a member of the low-density lipoprotein receptor gene family [56].

#### 2.3.2. CLU Nuclear Isoform (nCLU)

In contrast to sCLU, which originates from the full-length *CLU* mRNA, *nCLU* seems to be produced from the alternative messenger reported by Leskov et al. that is lacking the ER signal peptide containing exon 2. Consequently, lacking exon 2 has important consequences on CLU protein localization in the cell. In this alternative spliced transcript, the third ATG start site (+1487) is used for translation initiation and encodes for a shorter precursor protein (pnCLU), corresponding to amino acids 34 to 449 (Figure 2A), usually associated with cell death as its depletion was found to increase cell survival [38]. This precursor (pnCLU) appears as an intracellular unfolded 48 kDa protein. In response to stress condition, pnCLU is post-translationally modified (glycosylation), resulting in a mature pro-apoptotic nCLU of 55 kDa that does not undergo α/β cleavage and intensive glycosylation in Golgi [23,47,48,55] (Figure 3). nCLU contains three nuclear localization signals (NLSs at amino acids 74 to 81, 324 to 340 and 443 to 447) that relocate the nCLU protein from the cytoplasm to the nucleus following stresses (such as radiation) that induce pro-apoptotic effects [22,23,46,54]. Some data indicate that regulation of nCLU expression into the nuclei of irradiated cells occurs through both nuclear localization as well as nuclear export sequences (NESs), suggesting a very dynamic process in human cells in which the conversion of pnCLU to nCLU, as well as the continual export/import of nCLU protein, regulates the overall signal for cell death/survival. Furthermore, a mutation in the C-terminal NLS appears to abrogate the majority of nCLU function [23]. Further studies revealed that under stress condition, the CLU classical secretory pathway is evaded to reach the cytosol. This retro-translocation of CLU occurs through a mechanism similar to the ER-associated protein degradation pathway and involves passage through the Golgi apparatus [54].

**Figure 3 ijms-24-13182-f003:**
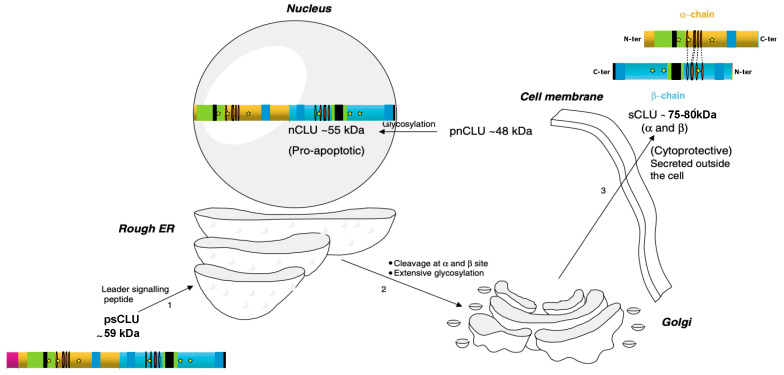
Generation of sCLU and nCLU. psCLU is transported into the rough ER by the leader signaling peptide (1), and then undergoes cleavage and extensive glycosylation while being transported to the Golgi apparatus (2). The result is a secreted protein of 75–80 kDa with five disulphide bonds between the α- and β-subunits that is secreted outside the cell (3). pnCLU does not undergo any cleavage or extensive glycosylation, and resides in the cytoplasm of unstressed cells. It becomes the mature form (55 kDa) once it is transported to the nucleus (see legend to Figure 2 for the color code) (adapted from ref [55]).

## 3. Clusterin Gene Regulation

It is now clear that the *CLU* gene encodes several transcripts, but very little is known about their regulation. Recent studies identified at least two distinct promoters that each impact the transcription of the *CLU* gene [45]. However, it is also not clear whether transcription of each mRNA is driven by a unique promoter or different promoters. The existence of several *CLU* mRNAs and the complexity of transcription regulation keep this question open. Moreover, the wide variation in the levels of *CLU* expression in different tissues suggests that it is tightly regulated, very tissue-specific, and finely tuned at transcriptional, translational and post-translational levels [57]. *CLU* is constitutively expressed in many healthy tissues but its upregulation is commonly associated with apoptosis, different pathologies and aging conditions. Therefore, it has been suggested that *CLU* isoforms’ differential expression may be controlled by diverse molecular machineries, signaling and epigenetic modifications (Figure 4). Gene-expression control is a complex and dynamic process involving many regulatory proteins that affect the expression of genes or transcripts in the cell at any given time [58,59]. In vertebrates, gene expression is usually regulated by the basal promoter (a short DNA region of up to 200 bp that overlaps the mRNA transcription start site) to which RNA polymerase II (RNA PolII) binds together with its accessory general transcription factors (such as TFIID) and regulatory transcription factors (TFs) (such as the ubiquitous TF Sp1). These regulatory proteins (including positive and negative TFs) influence the efficiency with which the entire *CLU* gene will be transcribed by RNA PolII, ultimately controlling protein synthesis in cells [60]. TFs, which can be tissue-specific or ubiquitous, are the essential molecular cell tools able to respond to environmental stress or stimuli changes (Figure 4). In addition, epigenetic modifications play a key role in gene expression regulation by affecting DNA accessibility and chromatin structure but do not involve DNA sequence changes (Figure 4) [61].

On the contrary, more than 6700 single nucleotide polymorphisms (SNPs) have been identified in the *CLU* gene. Most of them are synonymous variants but 367 are missense variants. Missense mutations involve a single base pair which is substituted by another and consequently alters the genetic code in such a way that it will encode an amino acid different from the usual and will alter the function of the protein. Fifteen SNPs generate frameshift variants, an insertion or a deletion of a number of base pairs that is not a multiple of three. Consequently, the mutation disrupts the triplet reading frame of a DNA sequence and usually creates an early termination (stop) codon, which results in a truncated (shorter-than-normal) protein product. Furthermore, there are 15 splice acceptors and 12 donor regions. Depending on the insertion or deletion, this type of variation can lead to frameshifts and missenses. Among the 6741 SNPs listed, only 14 have been identified in clinic [62]. Quite recently, one study aimed to assess the relationship between SNPs and predisposition or development of disabilities. Thus, some of them have been associated with neurodegenerative disorders [63,64]. In the eye, 5 SNPs (rs17466684, rs2279590, rs11136000, rs1532278 and rs3087554) have been identified in pseudoexfoliation syndrome during glaucoma and one has been associated with Fuchs’ endothelial dystrophy in cornea (rs17466684) [65,66,67,68,69,70].

Thus, a comprehensive understanding of health and diseases involves an evaluation of both DNA sequences and epigenetic modulation of gene expression.

### 3.1. Epigenetic Regulation

#### 3.1.1. Modification of Histones

Histones are proteins that regulate DNA compaction. The degree of compaction varies along the chromatin, being weak in euchromatin and more important in heterochromatin. This degree of DNA compaction is closely regulated by post-translational modification of the N-terminal ends of histones (which are subjected to many events such as phosphorylation, acetylation, methylation, ubiquitinations, sumoylation, etc.) that alter their interaction with DNA and transcriptional status [61,71]. Although modifications of histone tails are known to impact the chromatin structure, a complete understanding of the precise mechanisms explaining how histone end alterations influence DNA–histone interaction remains elusive. However, influences of some specific motifs have been well known, in particular modification on histone 3 (H3).

Histone acetylation generally facilitates transcription [61,72]. Thus, histones can lose their interaction with DNA and leave it more unwound and accessible to regulatory proteins (TFs) when their tails are acetylated at specific locations (such as the H3K9Ac (Histone3Lysine9acetylated) epigenetic mark) [72]. On the other hand, methylation of histones can lead to active and inactive transcription [73]. Specific genes are activated when tightly bound to histones post-translationnally modified on H3K4 (Histone3Lysine4) or H3K36 (Histone3Lysine36). Indeed, di- and tri-methylation (H3K4me2, H3K36me2, H3K4me3 and H3K36me3) of these histones facilitates access of TFs to certain DNA domains. Conversely, other types of histone methylation marks (such as H3K9me2/3 and H3K27me2/3) are rather associated with repression of gene transcription [61,74].

These outcomes are consistent with CLU studies carried out in human carcinoma [75,76,77], fibroblast [45], endothelial [78] and retinal pigment epithelial cells [79]. Results obtained in these studies demonstrated clearly that histone H3 hyperacetylation and loss of histone H3 methylation marks, by use of histone deacetylases (HDAC) and DNA methyltransferase (DNMT) inhibitors, respectively, promote a drastic increase in CLU expression [45,75,76,77,78,79,80]. More precisely, CLU expression has been linked with common histone marks. In this regard, studies clearly show that the histone marks H3K9me3 and H3K27me3 (both repressive marks) are more frequently observed in the *CLU* gene promoter relative to the H3K4me3 and H3K9AcS10P marks (both permissive marks) (Figure 4). This relative occupancy in the *CLU* promoter region by nucleosomes with the H3K9me3 and H3K27me3 histone marks is the probable cause of relatively low levels of nCLU expression in human colon cancer cell lines [76]. Furthermore, induction of *CLU* gene expression using epigenetic drugs (HDAC and DNMT inhibitors) is associated with increases in permissive histone marks (H3K4me2/3 and H3K9Ac, H3K9S10P) together with a decrease in the repressive marks (H3K27me3, H3K9me3) present on the *CLU* gene promoter [45,78,79] (Figure 4).

In neuroblastoma cells, *CLU* expression is repressed by the TF MYCN through the recruitment of HDACs and Polycomb group proteins. Absence of MYCN lifts repression and induces *CLU* expression. As with epigenetic drugs used in cancer cells, neuroblastoma cells lacking MYCN also have an enriched H3K4me2 permissive mark combined with a reduction in the repressive marks H3K27me3 and H3K9me3 in the *CLU* promoter region [80].

#### 3.1.2. DNA Methylation

Coordinated with histone modifications, DNA methylation is usually involved in differentiated mammalian cells to repress transcription. DNA methylation is a process by which methyl groups are added to the C5 position of Cys in CpG islands and thereby alter transcriptional activity without any change to the DNA sequence [61,73]. 

The presence of a G/C-rich, mini-CpG island in the 5′ end boundary of the conserved *CLU* promoter region suggests that DNA methylation contributes to regulation of *CLU* gene expression [12,18,45,77,81]. This observation is consistent with studies showing that promoter methylation is associated with weak CLU protein expression as shown in human prostate cells [45,77]. However, these data have been controversial, as neither Hellebrekers et al. nor Deb et al. could report any involvement of DNA methylation in *CLU* gene expression in either colon cancer cells or human umbilical vein endothelial cells (HUVEC) [76,78]. On the other hand, studies conducted on different tissues, including breast cancer cells [82], ovarian epithelial cancer cells [83] and rat fibroblast cells [81], confirmed the correlation between *CLU* expression and its promoter methylation status (Figure 4). Conversely, promoter demethylation experiments conducted in prostate cancer cells [45], endothelial cells [78] and retinal pigment epithelial cells [79] using epigenetic drugs significantly increased *CLU* gene transcription in these cells (Figure 4). Hence, it is not surprising to observe *CLU* promoter hypomethylation in human tumor cells that also correlate with a moderate/strong CLU expression (nCLU and sCLU) [76,77,82]. In rats, additional analyses reported a correlation between low levels of *CLU* promoter methylation and tissues (such as testis and epididymis) where CLU is constitutively expressed compared to those (such as lung, liver, spleen, kidney and prostate) with low/moderate CLU expression [84].

#### 3.1.3. MicroRNA

MicroRNAs (miRNAs) are small, single-stranded, non-coding RNAs (21 to 24 nucleotides) that post-transcriptionally regulate mRNA expression by base-pairing with complementary sequences within target mRNA transcripts. As a result, these mRNA molecules are silenced by cleavage of the mRNA strand, destabilization of the mRNA, or less efficient translation of the mRNA into proteins [61].

Studies describing the regulation of *CLU* gene expression by miRNA have emerged over the last few years. Works carried out in head and neck squamous cell carcinoma (HNSCC) revealed that the *CLU* mRNA is a specific and functional target of oncogenic miRNA-21 [85], while in neuroblastoma, *CLU* is strongly downregulated by MYCN, at least in part through transcriptional induction of the miR-17-92 microRNA cluster [86]. In lung adenocarcinoma, *CLU* is directly targeted by miR-378 [87]. In addition, the increased cell growth observed in HNSCCs also correlates with a reduced expression of the growth-suppressive CLU-1 isoform, a demonstrated target of miRNA-21 (over-expresed in these cells) [85]. *CLU* expression is also affected by miR-17-92 [86]. However, the direct targeting of the human *CLU* 3′UTR by miR-17-92 is still contested, as binding sites for miR-17-92 cluster members have been predicted by the Miranda algorithm, but these predictions could not be confirmed experimentally (using the luciferase sensor assay or gain-of-function microRNA mimic screens), suggesting that miR-17-92 does not directly target *CLU* but rather an upstream activator of *CLU* expression [86]. Evidence indicates that there are two alternative forms of CLU protein with opposite functions. HNSSCC cells preferentially express the growth-suppressive nCLU variant (also designated CLU-1), whose transcription is regulated by miRNA-21 [85]. Downregulation of nCLU expression was found to enhance tumor formation, therefore leading to the conclusion that CLU might also be a tumor and metastasis suppressor gene [86]. On the other hand, high levels of miRNA-378 inhibit sCLU expression and restore sensitivity to Cisplatin (an anticancerous drug) through regulation of sCLU-induced cell death signals [87], indicating that sCLU should be an oncogenic factor (Figure 4).

Overall, these results suggest a scenario in which oncogenic stimuli provoke chromatin rearrangements that result in: (i) suppression of nCLU expression, therefore restraining tumor proliferation and angiogenesis, or (ii) suppression of sCLU expression, involved in resistance to treatment. These results suggest again that an imbalance in the sCLU/nCLU ratio leads either to cell apoptosis or to cell survival.

### 3.2. Stress-Inducible Genes

Oxidative stress regulates a lot of signaling pathways leading to the activation of several TFs including HSF1, AP-1, NF-kB, p53 and STAT proteins [88,89,90] (Figure 4). Considering this modulation, the predictive combinatory presence of CLE (clusterin element, a 14 bp DNA region recognized by HSF-1), AP-1, NF-kB, p53 and STAT binding sites in the *CLU* promoter suggests that the *CLU* gene must act as an oxidative stress sensor [45,84,89,90,91]. This hypothesis is confirmed by studies showing that cellular damage (oxidative stress, hypoxia, radiation, heat shock and proteotoxic stress) influenced both *sCLU* and *nCLU* mRNA and protein expression in tissues such as breast, colon and prostate carcinoma, and endothelial cells [46,49,89,90,92,93], through stress-response elements such as hypoxia-response elements (HREs) [46] and both p53 [90,93] and HSF-1 [89] binding sites.

In human cells, sCLU and nCLU expression are induced by ionizing radiation. To define the function of CLU after cell-stress exposure is complexified by the existence of two isoforms of the protein with opposite functions since sCLU is commonly cytoprotective whereas nCLU exerts a pro-death function [23,94]. The majority of studies have reported that sCLU expression protects cells from a variety of agents that induce apoptosis (chemotherapeutic drugs, ionizing radiation, UV, H_2_O_2_, etc.), such as in human epidermoid cancer cells (where sCLU protects from H_2_O_2_, superoxide anion, hyperoxia and UVA) [49], normal human fibroblasts (where sCLU prevents cytotoxicity induced by ethanol, tert-butylhydroperoxide or UVB) [95], human prostate cancer cells (where oxidative stress-induced DNA damage is decreased by sCLU expression) [96], lung fibroblasts (from cigarette smoke oxidant) [97], in testis (from heat shock) [98], or in numerous studies that have reported cell resistance toward chemotherapeutic treatments due to CLU overexpression that is reversed by silencing CLU expression [24,25,26,50,99].

On the other hand, further in vivo studies have established that CLU accumulates in dying neurons of the brain and exacerbates neuronal damage in hypoxia-ischemia [100].

CLU also increases in human disease conditions where abnormal cell death or proliferation is observed. These include brain injury, chronic inflammation and neurodegenerative diseases (Alzheimer’s disease) [10,52], retinitis pigmentosa [101], various types of cancers (glioma and prostate, breast, etc.) [10,102], progressive glomerulopathy [6,103], atherosclerosis [104] and myocardial infarction [105].

The accumulation of CLU in human serum during atherogenesis, myocardial infarction and diabetes type II suggests that sCLU could be a vascular damage indicator. Its cytoprotective role became better understood in conditions such as atherosclerosis where sCLU accumulates in artery walls to protect against oxidative stresses induced by disease development [106]. However, in other tissues, the metabolic function of sCLU still remains controversial. For instance, increased CLU expression in Alzheimer’s disease is associated with amyloid beta peptide aggregation in senile plaques. In vitro, this overexpression of CLU prevents spontaneous aggregation of amyloid beta peptide and neurotoxicity, but pre-incubation of amyloid beta peptide with sCLU before adding to cell exacerbates cytotoxicity [107,108].

CLU is involved in various processes and tissues, sometimes with completely different functions. Some mechanisms are well described while others are still being studied. But, altogether, these results indicate that sCLU function may be tissue-dependent. Given its chaperone activity, sCLU induction could be a defense response in order to reduce cell damage and apoptosis induced by oxidative stress.

### 3.3. Transcription Factors

Initially, *CLU* mRNA was described in rats as a unique transcript of 1.9 kb with two in-frame ATG start sites (first in exon 2) and a promoter element containing a conserved palindrome (which is conserved in humans) from position −4220 to −4208, with several DNA-binding motifs potentially recognizable by various TFs. However, only a few studies have truly demonstrated interaction for a restricted number of these TFs with their corresponding elements in the *CLU* gene promoter [12,18,45] (Figure 4). Based on the distribution of *CLU* regulatory elements, the promoter was divided into three parts: region 1 (proximal) contains binding sites for the TFs AP-1, AP-2, Sp1 and NF-E1; region 2 is located upstream of region 1 and bears a cluster of response elements for CREB; and region 3 (distal) contains motifs for the TFs CTF, CRF, Hsp and GHF-1 [18]. In humans, the promoter (−4739 to −4029), which overlaps the isoform 1 transcription mRNA start site, contains a conventional TATAA box (−4160 à −4154) that directs recognition of the *CLU* promoter by the TBP-containing TFIID transcription factor [18]. Using bioinformatic tools, more than 700 potential TF binding sites can be found inside the 2000 bps of the *CLU* promoter. Among them, a 14 bp element (named CLE for clusterin element), which is highly preserved among vertebrate species and dissimilar from the heat-shock response element (HSE) by only one nucleotide, has been shown to bind HSF1 [89], therefore supporting a possible role of CLE as a cis-element for CLU transcription. More recently, further studies have revealed the existence of a second promoter region (−1350 to −749), lacking a TATAA box and located in the first *CLU* intron that overlaps the isoform 2 transcription start site. Bioinformatic tools predicted that this alternative promoter also contains response elements for TFs such as NFI and Sp1, which usually take part in the basic transcription machinery associated with these key regulators involved in development, cell proliferation and survival (NF-kB, MYCN and STAT binding elements) [45]. However, most studies on DNA–protein interactions related to *CLU* gene expression have been conducted in stress or pathological conditions, rather than in healthy cells, as discussed below (Section 4).

#### 3.3.1. AP-1

Early reports demonstrated the modulation of *CLU* gene expression during cell transformation, apoptosis and resistance to pharmacological treatments, suggesting that CLU participates in a survival response to apoptosis during tumorigenesis [10,25,26,99,109,110,111]. Consequently, some research groups investigated the impact of oncogenic transcription factors on *CLU* expression. The first evidence that *CLU* expression is modulated by oncogenic activity was reported in avian cells. In that study, the T64 gene (rat orthologue of the *CLU* gene) was activated by the retroviral oncogene v-src [112]. Subsequent investigations using a mutation of the AP-1 motif (TGACTCA) in the *CLU* promoter revealed that induction by the oncogenic kinases was dependent on the AP-1 binding site present in close proximity to the *CLU* transcription start site in quail neuroretina cells, and abolished CAT reporter activity [4]. The role of AP-1 (a complex containing the Jun and Fos oncoproteins) in regulating *CLU* expression (upregulation of transcript and protein levels) was later confirmed in rats [91,113]. Mutation of the AP-1 motif provided evidence that TGFβ positively modulates *CLU* expression via activation of an AP-1 site in the *CLU* promoter [91]. To further investigate the molecular mechanism involved in *CLU* regulation by TGFβ, the authors focused their interest on the AP-1 constituting subunits Jun, Fra and Fos and proposed that the mechanism of activation relies on the removal of the trans-repression effect of c-Fos by TGFβ [114]. However, the mechanism by which c-fos represses *CLU* expression has not yet been resolved. Gutacker et al. also reported that exposure of rat PC12 cells to NGF and EGF, respectively, induced neurite differentiation and cell proliferation, and *CLU* expression through AP-1 binding and activation [113]. Recently, analyses conducted in our laboratory revealed the presence of three regulatory regions that are important for proper expression of *CLU*: a basal proximal promoter and two more distal negative regulatory regions. Using electrophoretic mobility shift assays (EMSAs), we recently demonstrated the direct binding of the TFs AP-1 and Sp1/Sp3 to overlapping DNA binding sites in the basal *CLU* promoter. Interestingly, the expression of both these TFs in human corneal epithelial cells is reduced (at the protein level) during wound healing, thereby contributing to the extinction of *CLU* gene expression during that process [115].

#### 3.3.2. MYC and Ha-RAS

Claudia Koch-Brandt’s group studied the role of two classical proto-oncogenes, c-MYC and Ha-RAS, in the regulation of *CLU* expression. Overexpression of Ha-RAS, but not of c-MYC, in the rat embryo fibroblast cell line caused repression of *CLU* expression at the mRNA level [116]. Interestingly, RAS first induces *CLU* promoter deacetylation followed by methylation of a CpG island located in proximity to the transcription start site via MEK/ERK signaling, which also correlates with *CLU* reduction of expression [81]. Although it was initially reported that c-MYC does not regulate *CLU* expression, a more recent study revealed that ectopic levels of c-MYC could strongly repress the expression of *CLU* in murine colonocytes or human keratinocytes, thereby removing the CLU-dependent inhibition of their proliferation in vitro and during carcinogenesis in vivo [117].

Neuronal MYC (MYCN), a member of the MYC family, has been identified as a *CLU* negative regulator in human neuroblastoma cells [80,86]. MYCN amplification is characteristic of neuroblastoma and induces malignant processes leading to neuroblast formation. In tumors with amplified MYCN, *CLU* is strongly downregulated, in part through transcriptional induction of miR-17-92, and it was demonstrated that mice with deregulated *CLU* are more prone to the formation of neuroblastomas [86]. Their study provided evidence that MYCN could also directly repress *CLU* transcription through an E-box present in the 5′ flanking region of the *CLU* gene that is conserved in different mammalian species. Binding of MYCN on the *CLU* promoter is associated with bivalent epigenetic marks and the recruitment of chromatin remodeling proteins, such as histone deacetylases, that contribute to the repression of *CLU* gene expression [80]. The E-box also plays a role in binding the Twist transcription factor in prostate cancer cells. Indeed, Twist1 mediates basal and TGFβ–induced *CLU* expression by binding to E-boxes in the *CLU* gene distal promoter region. In addition, findings indicate that the Twist-CLU pathway is a mediator of TGFβ–induced epithelial–mesenchymal transition (EMT), migration and invasion in vitro and metastasis in vivo [118]. Twist is also required for IGF-I-mediated *CLU* expression and growth signaling in castration-resistant prostate cancer. IGF-I induces *CLU* expression via activation of the STAT3–Twist1 signaling pathway, leading to Twist1 binding to E-boxes on the *CLU* promoter [119].

As for MYC, MYB is a family of transcription factors that comprises the tissue specific c-MYB and A-MYB, and the ubiquitous B-MYB, a positive regulator of cell proliferation, differentiation and survival that is overexpressed in various types of human cancers including human neuroblastomas, suggesting that it functions as a proto-oncogene. To demonstrate such a function, B-MYB was experimentally overexpressed and the genes enhanced were identified. One of these B-MYB-regulated genes turned out to be *CLU*. B-MYB was subsequently shown to bind to and positively regulate *CLU* expression through an MYB-consensus sequence located in the 5′ *CLU* promoter region. It has also been demonstrated that CLU mediates, at least in part, the anti-apoptotic actions of B-MYB [120].

#### 3.3.3. NF-κB

The first evidence that *CLU* expression was under the regulatory influence of NF-κB has been provided by Li et al. [121]. A systematic analysis was conducted and the *CLU* gene was identified as one of the most highly NF-κB-activated genes in mouse embryo fibroblasts [121]. *CLU* induction by NF-kB signaling was later confirmed by using NF-kB inhibitors in glial and astrocyte cells stimulated by the bacterial lipopolysaccharide LPS [122]. Then, it was shown that, in humans, CLU negatively regulates NF-κB activity by stabilizing IκBs [123,124]. This led to the hypothesis that CLU participates in a negative loop that regulates NF-κB activity, which was later corroborated in rheumatoid arthritis, as the low CLU levels observed in the disease were shown to be associated with excessive NF-κB activation and exacerbated pro-inflammatory cytokines secretion [123]. However, this negative regulation is debated as another group demonstrated that CLU induces MMP9 activity through NF-kB translocation to the nucleus via IkBa degradation in macrophage cells [36]. Although *CLU* positive regulation by NF-kB was demonstrated, and a potential NF-kB response element was predicted to be present in the *CLU* promoter, binding of NFkB to this DNA target site has yet to be demonstrated.

#### 3.3.4. IGFR Pathway

sCLU induction by ionizing irradiation was first demonstrated by Yang et al. [22]. They later showed that irradiation leads to the activation of the IGFR (but not EGFR) signaling pathway and mediates the induction of sCLU. Notably, the IGFR receptor activated the Src/MEK/ERK signaling cascade, which triggered transactivation of early growth response-1 (Egr1) transcription factor that is required for upregulation of *CLU* mRNA [94].

#### 3.3.5. YB-1 and HIF-1a Signaling

Similarly, YB-1 (Y-box binding protein-1) (in response to endoplasmic reticulum stress) and HIF-1a (produced after exposure to hypoxia-mimetics) directly binds the *CLU* promoter in prostate cancer cells to stably regulate *CLU* expression [46,125].

#### 3.3.6. PAX6

PAX6 knockout in human corneal epithelial cells recently revealed its role in the regulation of *CLU* gene expression [126]. Indeed, a PAX6-dependent increase in *CLU* gene expression was observed in mice lacrimal glands [127].

#### 3.3.7. Nuclear Factor E2-Related Factor 2 (Nrf2)

In mouse hepatocytes, CLU expression signifcantly increases after exposure to hexavalent chromium (Cr(VI)), an environmental polluant that causes hepatocyte damage through the massive production of reactive oxygen species (ROS). Overexpression of CLU inhibits ROS production and alleviates Cr(VI)-induced liver injury through the protein kinase B (PKB/Akt)-Kelch-like ECH-associated protein 1 (Keap1)-Nrf2 signaling pathway, resulting in the release of Nrf2 into the nucleus. As Nrf2 has already been reported to regulate HO-1 expression (an antioxidant enzyme), the authors decided to focus their interest on this transcription factor. They demonstrated that the increase in Nrf2 is coordinated with a Cr(VI)-dependent induction of HO-1 via the Akt pathway [128].

#### 3.3.8. Hormonal Regulation of *CLU* Gene Expression

Studies conducted in hepatocytes have revealed that high glucose concentrations significantly increase CLU expression. However, the conventional promoter region of the *CLU* gene does not respond to glucose stimulation, while the first intronic region does [129]. That study reported the presence of a glucose response element (GlRE) consisting of two E-box motifs separated by five nucleotides and located in the first intron of the human clusterin gene, which can be affected by hyperglycemia and activates *CLU* expression through SREBP-1c (sterol regulatory element binding protein-1c) [129]. Studies performed in hepatocytes have also shown that SREBP-1c can activate *CLU* promoter and induce *CLU* expression in response to insulin. However, as with the glucose GlRE located in the first intronic region, the insulin response element was found to be a non-canonical E-box (NCE box) detected in exon 1 of the *CLU* gene [130].

Furthermore, a glucocorticoid/androgen-like response element has been identified in the 5′ region of the rat *CLU* promoter [18]. Activation of the ligand-bound glucocorticoid receptor increases expression of the anti-inflammatory cytokines while simultaneously inhibiting that of pro-inflammatory cytokines in order to attenuate the inflammatory response [131]. Furthermore, CLU expression is induced after treatment with glucocorticoids (dexamethasone). Treatment with antisense *CLU* oligonucleotides or antibody suggests CLU is responsible for the antiapoptic effect in human breast cancer. These results, together with the facts that glucocorticoids inhibit chemotherapy-induced cytotoxicity and that antisense oligonucleotides targeting CLU restore sensitivity to treatment, clearly show that glucocorticoids positively modulate the expression of CLU [25].

Interestingly, *CLU* mRNA levels were shown to increase dramatically in rat ventral prostate following castration [132]. This observation led to its designation as a testosterone-repressed transcript. Then, studies identified CLU as a stress response to apoptosis induced by castration rather than a protein repressed by testosterone [133]. This induction of CLU requires the androgen receptor (AR), and authors identified a putative androgen response element that is bound and transactivated by the AR. Studies carried out on human prostate tumor cells revealed that *CLU* transcripts are both expressed but differentially regulated by androgen treatment, leading to a reduction in the nCLU/sCLU expression ratio [134]. This androgen regulation of CLU may underline the cytoprotective role of androgens in normal prostate physiology as well as an anti-apoptotic function in prostate cancer progression. On the female side, rat endometrial carcinoma cells have evidenced that CLU expression is hormonally regulated [135]. Further studies have demonstrated that estrogens increase CLU expression in human endometrial cancer cells and enhance cell resistance to paclitaxel treatment [136]. Finally, thyroid hormones also downregulate CLU expression in human hepatocellular carcinoma cells after treatment by thyroid hormones [137].

## 4. Clusterin in Tissue Remodeling

As discussed above, CLU interacts with multiple other proteins. Additionally, it is not surprising to find it involved in diverse cellular functions, especially wound healing. Numerous roles have been assigned to CLU in regulating differentiation, proliferation, migration and survival of various cell types and cancer cells (Figure 5).

### 4.1. Differentiation

*CLU* is an early expressed gene found from 12.5 days old during mouse organogenesis, mostly in developing epithelia. In some tissues, proliferative and differentiative compartments can be distinguished. This is the case for the skin, teeth, and duodenum, where *CLU* expression is selectively localized within the differentiating cell layers. In addition, transient and localized *CLU* gene expression was detected in certain morphogenetically active epithelia, such as the lung during branching morphogenesis and the kidney during the process of polarization [16]. Based on their observations, it has been suggested that the expression of murine CLU is correlated with the cell remodelling or differentiation that occurs during this period of mouse embryonic development (from 12.5 to 18.5 days of gestation). However, the role of CLU in development remains unknown. As CLU is known for its aggregative function [1,138], one might predict that CLU expression in embryonic tissue development may confer adhesive properties and participate in cell movement needed for differentiation.

Other studies suggested a role for CLU in mesenchymal cell differentiation. Indeed, a few studies have reported that endogenous CLU supports the rapid formation of nodules and differentiation, characteristic of vascular abnormality, in monolayer cultures of vascular smooth muscle cells [139,140,141]. These observations suggest that CLU plays a critical role in regulating the phenotype of smooth muscle cells, with important implications for vascular diseases.

Additionally, there is growing evidence that CLU is involved in tissue remodelling in response to hormonal regulation and injury. In particular, it has been shown that CLU promotes proliferation and induces differentiation of non-differentiated duct cells into differentiated cells in pancreatectomized rats [142,143]. The same group, a few years later, completed this observation by showing that CLU induces differentiation of pancreatic duct cells into insulin-secreting cells (beta cells), suggesting that CLU may play essential roles in the neogenic regeneration of pancreatic tissue [142].

Altogether, the current literature demonstrates that CLU may have a potential role in regulating differentiation processes during embryogenesis and in response to injury (Figure 5).

### 4.2. Migration and Invasion

As previously discussed, in the context of vascular remodelling, CLU enhances nodule formation. Additional studies using Boyden chambers have demonstrated that, in such cells, CLU also promoted vascular smooth muscle cell (VSMC) migration [141]. However, contradicting these results, CLU has also been reported to inhibit migration of human aortic and murine femoral VSMCs through the repression of TNF-*α*-stimulated expression of ICAM-1, VCAM-1 and MMP-9 [144]. These proteins support cell mobility and could participate in the tissue remodelling occuring during pathologic processes such as atherosclerosis. Thus, regulation of their expression inevitably leads to inhibition of VSMC migration [145]. However, further investigation is required regarding the modulation of MMP9 by CLU, as some reports have demonstrated an inhibitory effect of CLU treatment on MMP9 expression and activity [145,146], while others have revealed a CLU-dependent protease activation [36,118,147]. In the tissue-engineered human cornea, CLU expression (mRNA and protein) decreases after damage [115], while MMP9 activity drastically increases in the central wound [148], therefore supporting that CLU might indeed negatively regulate MMP9 expression in the human cornea. Although the results obtained by the different groups initially appear incoherent, the regulation of MMP9 activity by CLU may depend on both the context and the function in the tissue of interest. For example, in cancer, CLU enhances MMP9 activation in macrophages [36] and breast cancer cells [147], which may contribute to the tissue reorganization into the tumor by serving as a modulator for ECM degradation and by promoting invasion of cancer cells. In addition, CLU also participates in the EMT transition [118]. In contrast, in preeclampsia (PE) pregnancy disease, CLU expression is increased before the clinical syndrome occurs and the CLU level was positively related to the severity of PE. In this pathology, CLU inhibited the expression of both MMP-9 and vimentin, whereas it enhanced that of E-cadherin, to inhibit EMT of trophoblast cells [149]. As a consequence, migration and invasion is reduced, which may contribute to the abnormal placental development that is typical of preeclampsia. Collectively, these results indicate that when the biological processes are deflected as in cancer or preecalmpsia, CLU may regulate either the aggressive behaviour of cancer cells or the EMT of trophoblast cells via regulation of MMP9 activity. On the other hand, CLU maintains epithelial homeostasis during inflammation, at least in part by CLU interaction with MMP9 [146], and protects against neointimal hyperplasia development during vascular injury through its antiproliferative and antimigratory effects on VSMCs [145,150] driven by an increased autophagy [150]. In normal vessels, CLU overexpression represses VCAM, ICAM and MMP9 expression, leading to inhibition of both VSMC migration and DNA synthesis, whereas it promotes reendothelialization after wire injury of the femoral arteries [145]. 

Apart from MMP proteins, other observations have favored pro-migratory roles for CLU. Results obtained in human pulmonary artery smooth muscle cells, in which sCLU is induced after pulmonary arterial hypertension, have demonstrated that CLU promotes proliferation, migration and resistance to apoptosis in part via both the Erk1/2 and Akt signaling pathways. Thus, these results demonstrate that sCLU plays central roles in pulmonary vascular remodelling, and its upregulation in lung tissues is obviously linked with the pathological progression in rats [151].

In immune cells, CLU has been associated with increased chemotactic migration of human monocytes and murine peritoneal macrophages through a CLU-dependent induction mechanism [152,153]. Administration of pertussis toxin, Gβγ inhibitor or MAPK-specific inhibitor (PI3K, PLC) suppressed CLU-induced migration, suggesting that it was G-protein-coupled, Gβγ-pathway-dependent [152]. More precisely, and in accordance with previous studies [152], involvement of the MAPK pathway has been demonstrated. Thus, CLU can induce chemotactic migration and the secretion of TNFα via ERK-, JNK- and PI3K/Akt-dependent pathways [36]. In addition, Shim et al. also suggested that CLU regulates expression of chemotactic cytokines, such as monocyte chemotactic protein-1 (MCP-1), macrophage inflammatory protein-1b (MIP-1b) and regulated upon activation and normal T cell expressed and secreted (RANTES) in macrophages [153]. Evidence was also provided that regulation of MMP9 activity by CLU was also dependent on signaling through the ERK, JNK and PI3K pathways [36]. Together, these results suggest that CLU involvement in tissue remodelling relies on ECM degradation and macrophage-dependent inflammation via activation of the MAPK signaling pathway [36,153].

In summary, reports about a role for CLU during vascular remodeling appear to be controversial and tissue-specific. However, that CLU enhances migration of mesenchymal cells and serves as a molecular bridge between inflammation and remodelling via recruitment of immune cells, including macrophages, is now a well established fact.

### 4.3. Proliferation

Several lines of evidence suggest a major role of CLU in tissue repair through the induction of proliferation of various types of cells after injury.

This is notably the case for renal tissue repair after ischemia-reperfusion injury. Indeed, in CLU-deficient mice, the progression of spontaneous renal regeneration is disrupted. This is associated with a decreased proliferation and survival of renal cells. In vitro data strongly suggest that CLU also functions as a mediator of proliferation, but not migration, in tubular epithelial cells [154]. Similar results have also been observed in primary astrocytes. Indeed, CLU expression is upregulated in brain injury that is typically observed in neurodegenerative diseases. In such instances, CLU induces primary astrocytes proliferation in in vitro culture via ERK1/2 signaling, suggesting a potential role in the astrogliosis occuring during the physiopathology development [155]. In the pancreas, Kim et al. noted a post-injury (induction of diabetes or pancreatectomy) regeneration of the pancreas associated with enhanced proliferation and differentiation of MIN6 insulinoma cells and primary pancreatic ducts cells through upregulation of CLU [142,156].

Contrary to the previous tissues in which the physiopathological condition induced proliferation via CLU upregulation, in injury-induced neointimal hyperplasia, sCLU overexpression is clearly observed but its functions still remain controversial. In some studies, sCLU upregulation enhances VSMC proliferation by promoting cell cycle pursuit, as demonstrated by sCLU silencing experiments that also induce G1 cell arrest, therefore suggesting that sCLU would contribute to restenosis after vascular injury [157]. However, these results differ considerably from those of Han-Jong et al. in which sCLU rather suppresses VSMC proliferation in vitro by inhibiting DNA synthesis (inducing G1 arrest of cells) without any induction of apoptosis. In that study, CLU deficiency enhanced VSMC proliferation in vitro and accelerated neointimal hyperplasia in vivo, but concomitantly impaired reendothelialization in murine femoral arteries in vivo and in human umbilical vein endothelial cells (HUVECs) in vitro, suggesting a protective role of CLU during vascular injury rather than a causative role in the pathogenesis of neointimal hyperplasia [145].

Although it is not clear to what extent the expression of CLU acts on the cell cycle, it was shown that sCLU is cell-cycle-related in human dermal fibroblasts, with its accumulation occuring during the G0 phase [158]. CLU overexpression in immortalized human prostate epithelial cells resulted in an increased accumulation of cells at the G0/G1 phases of the cell cycle, accompanied by a slowdown in cell cycle progression and a reduction in DNA synthesis [159].

In pulmonary fibrosis, CLU, which is normally localized in fibroblasts, is downregulated, at least in part due to TGF-B1 induction and reduced fibroblast proliferation. This process represents an appropriate but insufficient response in order to limit fibroblast proliferation associated with disease progression [160].

In summary, these data strongly support a major role for CLU in the regulation of cell proliferation. According to the tissue/cell, CLU expression can either enhance or repress proliferation and protect or aggravate the pathology, evidencing an apparent tissue- and cell-dependent CLU regulation (Figure 5).

### 4.4. Apoptosis vs. Cytoprotection

Numerous studies have reported a correlation between CLU overexpression and pro-death signals in cells, while others rather observed cytoprotective functions of CLU proteins. Although the structure/function relationship is not completely understood, nCLU, located in the nucleus, is commonly associated with apoptotic functions, wherein it inhibits cell growth and survival, whereas sCLU is a pro-survival secreted signal overexpressed in cancer and causes treatment resistance and protection against cytotoxic agents inducing apoptosis.

It has been suggested that CLU isoforms reside in the nuclear and cytosolic compartments of human cell types after damage, while in viable cells CLU is specifically cytosolic and then secreted. Several mechanisms have been proposed to explain this atypical location of CLU proteins. Results suggest that an inactive precursor of nCLU (pnCLU) exists in the cytoplasm of non-irradiated cells that translocates into the nucleus (through a nuclear localization signal (NLS) present on the protein) upon ionizing irradiation (IR) [23]. Silencing or overexpression of nCLU confirmed nCLU involvement in IR-induced apoptosis and identified it as a pro-death factor expressed in the cell nucleus [22,38]. However, its function is not fully understood. Interestingly, CLU has been found to be associated with the DNA-dependent protein kinase (DNA-PK) double-strand break (DSB) repair complex. Therefore, nCLU (but not sCLU) binds to Ku70, forming a trimeric protein complex with Ku80 (two components of the DNA-PK complex) involved in nonhomologous DNA DSB repair. Overexpression of CLU has been shown to reduce the binding activity of Ku70/Ku80 to DNA ends in whole-cell extracts, suggesting that nCLU may directly affect DSB repair [22,34]. More recent data indicate that Ku70 and sCLU play an important role in binding Bcl-2-associated X protein (Bax) to control cell death [21,161]. Interaction between Ku70 and Bax protects cells from the Bax-mediated cell death response [161]. Whereas sCLU expression stabilizes the Ku70-Bax complex, nCLU modulates the complex and releases Bax from the cytoprotective Ku70–Bax complex leading to Bax relocation to mitochondria and apoptosis [38,161,162]. Furthermore, nCLU has also been shown to interact with bcl-XL (a member of the bcl-2 family of proteins involved in heterodimer formation with Bax) via its C-terminal domain. This interaction, besides sequestering bcl-XL to the nucleus, also causes the release of Bax from n-CLU and promotes apoptosis [33]. Irradiation is associated with an increased expression of sCLU. However, and contrary to nCLU, sCLU silencing promotes cell death. Considering that IR enhanced apoptosis, this result could be surprising at first. However, further analysis revealed that IR-induction of sCLU is linked to p53 upregulation and confirmed a role for this tumor suppressor protein in the transcriptional repression of sCLU, suggesting that p53-dependent repression of sCLU protein levels may lead to cell death induced by IR or cytotoxic agents [90]. In contrast, sCLU depletion activates p53 and decreases bcl-2 expression, triggering apoptosis [157,161]. However, p53 involvement in the induction of bcl-2 remains controversial, as bcl-2 expression can also be induced through a p53-independent mechanism [161,163]. From a molecular point of view, results obtained in cells resistant to hypoxia-induced cell death compared to sensitive cells demonstrated that nCLU expression and its associated reduced cell viability are also dependent on CLU promoter demethylation [46] (Figure 5).

Most of the evidences for a cytoprotective function of CLU has resulted from analyses conducted in cancer cells (refer to Section 5 below). However, some studies conducted under injury conditions (stress or pathological) have shown the importance of sCLU in cell survival outside the metastatic context.

The cytoprotective function of sCLU expression has been observed in response to stress, including heat shock and oxidative stresses. Anti-sense transfectant strategies restored cell death sensitivity to stress in human epidermoid carcinoma [49], while sCLU overexpression enhanced cell viability in human retina during retinal development and in the pathogenesis of age-related macular degeneration (AMD) [164,165] (Figure 5).

Additional data also suggest that sCLU acts as an extracellular HSP-like chaperone, scavenging denatured proteins outside cells by binding to hydrophobic regions of partially unfolded proteins following specific stress-induced injury and via an ATP-independent mechanism [14,27]. It inhibits protein aggregation and precipitation, which would otherwise be caused by physical or chemical stresses (e.g., heat, oxidative reduction) [28] (Figure 5).

Following renal (ischemia-reperfusion) injuries (IRIs), upregulation of sCLU is observed and is associated with cytoprotective functions, as well as an upregulation of a panel of genes that could positively regulate cell cycle progression and DNA damage repair, which might potentially promote cell proliferation. These data suggest that sCLU is required for renal tissue regeneration in the kidney repair phase after IRI [154].

As alluded previously, sCLU has been suggested to play an important role in pulmonary vascular remodeling in response to injury, wherein sCLU induces resistance to apoptosis in human PASMCs, potentially mediated through ERK1/2 and Akt signaling pathways. That report also provides evidence for the reverse effect, as CLU silencing was shown to spontaneously induce apoptosis [151]. The analysis of primary lung fibroblasts revealed a downregulation of CLU expression in fibrotic lung fibroblasts compared to control lung fibroblasts. ShRNA-mediated downregulation of CLU sensitized cells to apoptosis, representing a way to limit fibroproliferation associated with this pathology [160].

In the liver, CLU is expressed in hepatocytes, and its upregulation is observed during growth and regeneration. In the liver, response to injury by the hepatomitogen cyproterone acetate (CPA) can be divided into three phases: (i) the initial growth, which is due to hyperplasia, (ii) the plateau of liver enlargement, representing a new homeostatic set point and resulting from the feedback inhibition of cell proliferation, and (iii) the regression of liver size and DNA content after the cessation of CPA treatment, which includes the selective death of hepatocytes by apoptosis [166]. The steady state level of *CLU* mRNA was found to exhibit a corresponding triphasic pattern; the level increased during the initial growth phase of the liver and during liver regression, and then decreased when liver size reached its plateau [166]. Although no distinction was made in that study between nCLU and sCLU expression, which probably explains the triphasic pattern observed, it remains clear that CLU expression is associated with pro-survival or pro-apoptotic processes.

In fibroblast-like synoviocytes (FLSs), the effects of sCLU are in contrast to those observed previously; CLU is downregulated in rheumatoid arthritis, with a decrease in extracellular CLU and barely detectable intracellular forms of the protein [123]. That study showed that CLU overexpression inhibited NF-kB activation and induced apoptosis within 24 h, while CLU silencing enhanced production of the pro-inflammatory cytokines (IL-6 and IL-8), suggesting that weak CLU expression in FLSs may enhance NF-κB activation and the survival of synoviocytes [123].

## 5. Clusterin and Cancer

CLU has become a focal topic in the scientific community due to its crucial role in numerous biological processes such as cell survival, apoptosis, epithelial–mesenchymal transition (EMT), metastasis and chemoresistance (Figure 5). The acquired knowledge about this multifaceted protein has been made through extensive research within the field of oncology. Overexpression of CLU has been observed in various human cancers, making it a potential therapeutic target and a valuable research opportunity for understanding the underlying mechanisms of tumor progression.

### 5.1. Clusterin Isoforms’ Function in Cancer Progression

As stated earlier, the *CLU* gene produces at least two isoforms of clusterin: nuclear (nCLU, isoform 1) and secreted (sCLU, isoform 2) [39]. Each isoform has distinct functions; nCLU triggers cell death (apoptosis), while sCLU inhibits it. Additionally, sCLU is known to be cytoprotective due to its chaperone-like activity [167]. The proportion of these two isoforms, along with an increasing concentration of sCLU, may explain the development of more aggressive and metastatic cancers. In a prostate cancer cell line (LNCap), an increase in the expression ratio of nCLU/sCLU, through on-demand alternative splicing, showed increased sensitivity to radiotherapy and chemotherapy [167]. In human colon carcinoma, the translocation of CLU from the nucleus to the cytoplasm has been directly linked to tumor progression [168]. In either cancerous or normal epithelial prostate cells, CLU’s translocation from the cytoplasm to the nucleus causes cell cycle arrest and apoptosis [169]. In breast carcinoma, the expression of clusterin increases as the tumor progresses. The use of immunohistochemical staining in breast lesion samples revealed a rise in CLU expression in benign lesions (19%), atypical hyperplasia (47%), in situ carcinomas (49%), invasive carcinomas (53%) and metastatic breast lesions (80%). The staining was mainly observed in the cytoplasm and was associated with a lower apoptotic index in CLU-positive tumors [110]. Both isoforms of CLU are distinctly regulated by androgens; isoform 2 being upregulated while isoform 1 being downregulated, thereby altering the nCLU/sCLU ratio [134,170]. Androgen receptors (ARs) also play a role in the regulation of *CLU* gene expression by interacting with androgen responsive elements located in the first intron of the *CLU* gene; this interaction results in an increase in CLU expression [134]. In breast carcinomas, overexpression of clusterin is significantly associated with a negative expression of estrogen and progesterone receptor status [110].

Huang and his colleagues found that cytoplasmic precursor CLU (psCLU) is downregulated in lung cancer tissue, with a slight decrease in the secretory CLU proteins. Notably, an increase in CLU strongly hinders the movement, invasion and spread of lung cancer cells, while reducing CLU has the opposite effect. However, secretory CLU does not seem to have the same effect against metastasis. The reduction of CLU is strongly linked to tumor metastasis. Additionally, the overexpression of CLU demonstrated an anti-metastatic effect both in vitro and in vivo. The cytoplasmic precursor of CLU binds to ROCK1, reducing the phosphorylation of ERK1/2 by suppressing the kinase activity of ROCK1 and thereby disrupting the connection between ROCK1 and ERK, therefore hindering ERK activity and reducing lung cancer’s ability to invade. At the same time, the presence of CLU was found to be significantly lower in areas of lung cancer that had spread to the bone, compared to tumors beneath the skin in mouse models, and was scarcely found in the bone metastasis sites in human lung cancer patients when contrasted with the primary location [171].

### 5.2. Tumorigenesis 

In normal cells, CLU expression is low and greatly increases in response to stress factors. This increase is associated with the activation of transcription that induces clusterin mRNA [49,172]. When exposed to stress, sCLU (cytoprotective isoform) protects the cell through its extracellular chaperone activity, thereby preventing apoptosis [167]. sCLU confers resistance to cytotoxic agents, and its depletion in non-stressed human cancer cells results in changes in the balance of pro-apoptotic to anti-apoptotic proteins in the Bcl-2 family, activates p53-dependent cell cycle, slows down growth and increases mitochondrial apoptosis. sCLU protects cells by binding to the Ku70-bax complex, functioning as a retaining factor for bax in the cytosol, and thus inhibiting bax’s pro-apoptotic role. The overexpression of sCLU stabilizes the Ku70-bax complex, promoting tumorigenesis by preventing the translocation of bax to the mitochondria, which in turn inhibits apoptosis. CLU inhibition weakens the Ku70-bax complex, increases translocation of bax to the mitochondria, releases cytochrome c and activates caspase 9, which initiates apoptosis [161]. Therefore, high levels of sCLU may increase tumor growth by impeding cell death [21,161,162]. Other different signaling pathways regulate CLU’s anti-apoptotic function, such as B-MYB transcription factor, or by Akt phosphorylation and IGF-1, which promotes activation of the PI3K/Akt pathway [120,173]. Meanwhile, sCLU has been reported to activate the PI3K/Akt axis, thereby leading to phosphorylation of GSK-3β and inhibition of apoptosis [24]. GRP78 also promotes CLU to cytoprotect cancerous cells in hepatocellular carcinoma (HCC) [174]. On the other hand, nCLU (pro-apoptotic isoform) is involved in inducing apoptosis [23]. In breast cancer, nCLU binds with cytosolic Ku70 which results in cell death after ionizing radiation [22,175]. In prostate cancer, nCLU interacts with Bcl-XL, reducing the formation of Ku70-bax complex. This leads to the release of bax, causing cell death through the activation of cytochrome c and caspase-3 release [33]. nCLU also arrests prostate cancer cells in G-2M by downregulation of the mitotic complex cyclin B1 and CDK1, which induces cell death [169]. CLU function in tumorigenesis may be related to the ratio of its isoform production [168]. Essentially, the ratio of both isoforms is key for regulating the apoptotic and anti-apoptotic functions of the protein. Overexpression of CLU promotes tumorigenesis and chemoresistance in multiple cancers like breast and prostate cancer [176,177].

### 5.3. Epithelial to Mesenchymal Transition and Metastasis 

The EMT is a biological process that results in significant changes in the organizational structure of cells from epithelial to mesenchymal type. This transition results in a suite of morphological and functional alterations, including the remodeling of the cytoskeleton, loss of cell polarity and adhesion, alterations in cell–matrix adhesion, acquisition of a fibroblast-like morphology, increased motility and the ability to invade the basement membrane. These cellular changes confer upon cells a set of invasive and metastatic properties [178]. This process leads to metastasis and can be summarized as EMT-mediated invasion, degradation of basement membrane, intravasation through vascular or lymphatic vessels, extravasation into another organ and proliferation of a secondary tumor [178]. The process is regulated by different signaling transduction proteins such as matrix metalloproteinases (MMPs), E-cadherin and transforming growth factor (TGF). Recent studies have shown the pivotal role of CLU in regulating EMT, invasion and metastasis via different pathways. The activity of MMP-9, which is regulated and promoted by CLU, was discovered to be reliant on the activation of ERK1/2 and PI3K/Akt [36,179]. Clusterin also facilitated the movement of NF-κB p65 into the nucleus and caused the breakdown and phosphorylation of IκB-α, which was crucial for the expression of MMP-9. Given that NF-κB is a key player in inflammation, clusterin may act as a molecular connection between inflammation and cancer by elevating both NF-κB and MMP-9. NF-κB increases MMP-2 and MMP-9 expression, thus promoting metastasis [36]. In clear cell renal carcinoma (CCRC), the overexpression of CLU regulates tumor progression and metastasis of human CRCC cells via modulation of ERK1/2 signaling and MMP-9 expression [180]. The overexpression of CLU in epithelial ovarian cancer cells promotes tumor angiogenesis via VEGF secretion [181].

CLU expression has been correlated with metastasis in different types of cancers such as nasopharyngeal carcinoma [182], breast cancer [183], hepatocellular carcinoma [184], colon cancer [185] and prostate cancer [186]. In nasopharyngeal carcinoma, *N,N*′-dinitrosopiperaxine (DNP), a carcinogenic factor, upregulates CLU, thereby inducing MMP-9 and VEGF expression, which promote metastasis [182]. In breast cancer, CLU has been shown to enhance the effects of eHsp90α in activating important protein families, including Akt, ERK and NF-κB. This activation promotes EMT, migration and in vivo tumor metastasis [183]. Platelets increase colon cancer invasion by activating the p38MAPK pathway and upregulating MMP-9. This process is facilitated by clusterin and thrombospondin 1 (TSP1). In an in vitro study, it was demonstrated that miRNA-217-5p regulates invasion and migration in prostate cancer by targeting CLU [186]. Yang and his team concentrated their study on patients with pancreatic ductal adenocarcinoma (PDAC). Within this patient group, they discovered an irregularity in the regulation of CLU, specifically in the subset of patients exhibiting the most aggressive form of the disease. Hepatocyte nuclear factor 1 b (HNF1B) was found to upregulate CLU, and a diminished expression of both HNF1B and CLU correlated with worse patient survival. In summary, the study indicates that the HNF1B/CLU pathway serves to hinder the advancement of pancreatic cancer. The HNF1B/CLU axis’s dysregulation contributes to the advancement of PDAC. A reduction in CLU levels leads to increased cell proliferation and EMT, along with decreased sensitivity to the chemotherapy drug gemcitabine. These factors collectively result in accelerated disease progression and poorer outcomes for patients [187].

### 5.4. CLU Oncogenic Biomarker

In the medical field, biomarkers provide valuable information regarding a patient’s health and disease progression. These biomarkers range from straightforward protein levels in the blood to genetic mutations that are often used for diagnosis, prognosis, monitoring and treatment. Many studies have focussed on the use of CLU as a potential biomarker in cancer. This is of high interest since CLU is overexpressed in many cancers. The soluble level of CLU was evaluated as a biomarker in different cancers such as hepatocellular carcinoma [188,189] and colorectal cancer [190]. The results indicated a significantly high level of CLU in cancer patients in comparison to the control group.

In a recent meta-analysis, Gao and his colleagues studied the diagnostic accuracy of CLU in patients with hepatocellular carcinoma. This selected study evaluated the utility of CLU in distinguishing between patients with hepatocellular carcinoma and non-hepatocellular carcinoma conditions, such as liver cirrhosis, chronic hepatitis and other benign liver diseases. It was observed that the serum levels of CLU were elavated in the hepatocellular carcinoma group compared to the liver cirrhosis group, suggesting an involvement of CLU in the process of carcinogenesis. As a positive control, alpha-fetoprotein (AFP), a protein produced by certain types of cancer cells, including hepatocellular carcinoma, was utilized. AFP levels in the blood can be measured through a blood test, and elevated levels of AFP along with other imaging technologies can suggest hepatocellular carcinoma. The study compared the sensitivity, specificity, diagnostic odds ratio (DOR) and area under the curve (AUC) for both CLU and AFP. Compared to AFP, CLU proves to be a superior biomarker for the diagnosis of hepatocellular carcinoma. The combined use of CLU and AFP significantly enhances the diagnostic performance, resulting in an even higher sensitivity, DOR and AUC [191].

Since more research is necessary to determine an exact value for the CLU level associated with cancer, for now it can be used as a risk assessment tool [192].

### 5.5. Chemoresistance and Chemosensitivity with Clusterin 

Chemoresistance refers to the ability of cancer cells to resist the effects of chemotherapy drugs. This can occur through a variety of mechanisms, such as changes in genes or proteins, alteration of the drug metabolism, drug resistance, suppression of apoptosis, development of drug efflux pumps and upregulation of DNA repair [193]. Chaperone proteins play a crucial role in cancer therapy resistance by allowing and facilitating the growth of malignant tumors. These proteins help maintain the stability and proper folding of cellular proteins, thereby protecting drug resistant proteins [194].

Many treatment options used to eliminate cancerous cells induce a stress response from the human body. One of these responses is the overexpression of sCLU, a stress-activated cytoprotective chaperone, when exposed to chemotherapy [195]. Radiotherapy and chemotherapy are cytotoxic to the tumor cells. Clusterin plays a crucial role in chemoresistance by interacting with activated Bax, blocking the release of cytochrome c and the process of apoptosis [21].

The downregulation of CLU expression found normally in testicular seminoma results in its high sensitivity to radiotherapy and chemotherapy [196]. In gastric cancer cells, the overexpression of sCLU leads to downregulation of miR-195-5p, a microRNA molecule, which increases chemoresistance [197].

sCLU is highly overexpressed in hepatocellular carcinoma (HCC) and contributes to oxaliplatin resistance by downregulating the expression of Gadd45a and activating the PI3K/Akt pathway [198]. Also, silencing CLU in HepG2/ADM cells of HCC restores chemosensitivity to drugs like irinotecan, gemcitabine, cisplatinum and doxorubicin. With doxorubicin, *CLU*’s silencing inhibits the drug efflux pumps and therefore promotes apoptosis [199]. In pancreatic cancer, the inhibition of CLU decreases NF-κB/bcl-2 pathway activity, thereby increasing the apoptotic effect of gemcitabine chemotherapy [200].

Based on current evidence, it has been shown that inhibiting CLU leads to an enhancement of chemosensitivity, through either siRNA genetic inhibition or antisense inhibition [201]. Custirsen (OGX-011) is a second-generation antisense oligonucleotide (ASO) (2′-methoxyethyl modified phosphorothioate antisense oligonucleotide) that blocks *CLU* mRNA, resulting in a decreased clusterin synthesis, decreased cell proliferation, increased sensitivity to chemotherapeutic drugs and improved clinical outcomes for patients with castration-resistant prostatic cancer (CRPC).

The first-generation of ASOs had a short tissue half-life, which required uninterrupted IV infusion, which caused significant constraints. To increase the stability and effectiveness of ASOs, researchers have attempted to modify the phosphodiester linkage, the heterocycle, or the sugar components of the ASO molecule [202]. Custirsen has the 2′MOE modification with four 2′MOE-modified nucleosides at the 3′side, four 2′MOE-modified nucleosides at the 5′ side and thirteen 2′-deoxyribonucleosides in between (often referred to as a 4-13-4 MOE gapmer) [202].

Custirsen has demonstrated promising results in preclinical efficacy studies. It has been shown to augment the therapeutic benefits of existing treatments such as hormonal therapy, chemotherapy and radiating therapy in multiple preclinical tumor models. They included a wide range of malignancies such as prostate cancer, breast cancer, non-small-cell lung cancer, bladder cancer and kidney cancer [203].

In non-small-cell lung cancer (NSCLC), the addition of miR-378, a microRNA, to patient samples downregulated sCLU and overcame chemoresistance to cisplatin [87]. In head and neck squamous cell carcinoma (HNSCC), the protein CLU is targeted by an overexpressed oncogenic microRNA called miRNA-21. CLU has a growth-suppressive function in HNSCC, and this function is targeted by miRNA-21. Therefore, by targeting and inhibiting *CLU*, miRNA-21 can promote the growth and progression of HNSCC [85].

The combination of sorafenib and OGX-011 was found to enhance the cytotoxic effect of sorafenib on renal cell carcinoma (RCC) by stimulating apoptosis, in both in vitro and in vivo models. This effect was achieved through the downregulation of phosphorylated Akt and p44/42 mitogen-activated protein kinase. In in vivo models, the combined treatment significantly decreased the tumor volume of ACHN, a human renal cell carcinoma cell line, compared to the control group treated with oligodeoxynucleotide only [204].

The use of 100 µM of ascorbate was found to induce apoptosis in A2058 melanoma cells. RNA-sequencing analysis revealed a decrease in the expression of cytoplasmic CLU (an anti-apoptotic protein) and a sustained expression of nCLU (a pro-apoptotic protein) [205].

Among pharmacological drugs used in cancer treatment, Zoledronic acid (ZOL) is utilized as an adjunctive therapy in chemotherapy to limit osteolysis in osteosarcoma. Yet, it also causes an increase in the levels of heat shock proteins, such as Hsp27 and CLU, which have the potential to promote the survival of tumor cells and resistance to treatment. Therefore, the inhibition of CLU by OGX-11 synergistically potentiates ZOL’s cytotoxic effect in cancerous cells and delays progression of osteosarcoma [206]. It has also been shown that the depletion of sCLU alters ratios of pro- and anti-apoptotic Bcl-2 proteins, retards cell growth in the G1-S phase of the cell cycle via p53 activation and increases mitochondrial apoptosis [161].

Castration-resistant prostate cancer (CRPC) refers to a stage of advanced prostate cancer where the cancer cells continue to grow and spread despite the suppression, by medical or surgical castration, of male hormones (androgens) [207]. It occurs when the cancer cells find ways to adapt and grow despite the decrease in androgen levels resulting from hormone therapy [208]. The combined administration of MDV3100, an androgen-receptor antagonist, and OGX-011 exhibits a synergistic effect by targeting the androgen receptor (AR) and clusterin (CLU), respectively, leading to increased rates of programmed cell death compared to the use of MDV3100 or OGX-011 alone. This combination also resulted in delaying the progression of CRPC LNCaP tumors and prostate-specific antigen (PSA) levels in vivo [209]. The combination of Hsp90 inhibitors and OGX-011 in xenograft models of human CRPC also reduces the heat shock response triggered by Hsp90 monotherapy, inhibits tumor proliferation by 80%, improves survival and impedes CRPC progression [207].

In conclusion, custirsen (OGX-011) shows promising potential in the treatment of castration-resistant prostate cancer. It has demonstrated synergistic effects when combined with other therapeutic agents such as hormone therapy, chemotherapy and Hsp90 inhibitors. Further research is needed to fully explore its potential in combating cancer progression and improving clinical outcomes.

### 5.6. Clinical Trials of CLU Inhibitors

In a phase I/II trial, the combination of gemcitabine and platinum with custirsen (OGX-011) was studied in patients with stage IIB/IV non-small-cell lung cancer (NSCLC). The study found that 95% of patients experienced a decrease in CLU serum levels. Patients whose minimum median CLU level was below 38 μg/mL during treatment had a higher mean survival of 27.1 months compared to 16.1 months for those who did not reach this threshold. A larger phase III randomized trial is necessary to validate the benefit of custirsen [210].

In a phase II trial, patients with metastatic breast cancer tolerated the combination of OGX-011 and docetaxel, which showed some clinical activity. However, the number of responses was not enough to fulfill the requirements for proceeding to the second stage of accrual [211].

In a randomized phase II trial, a chemotherapy-resistant metastatic CRPC population received either docetaxel or mitoxantrone, both combined with custirsen. The aim was to evaluate the hypothesis that custirsen could either reverse docetaxel resistance or enhance mitoxantrone efficacy. The use of the combined treatment following the initial docetaxel therapy resulted in significant pain relief and revealed a connection between the level of serum CLU and the survival of patients [212]. During the randomized phase II trial, the combined treatment of docetaxel and prednisone was well received by patients and was found to be linked to increased survival rates in metastatic CRPC [213].

In the randomized international phase III trial, the inclusion of custirsen in the treatment of cabazitaxel and prednisone did not result in any improvement in the survival of men with metastatic castration-resistant prostate cancer [214]. As a result, the standard treatment for patients with this type of cancer that has progressed after docetaxel chemotherapy remains prednisone and cabazitaxel.

In conclusion, the combination of custirsen with different chemotherapy regimens showed promising results in various early phase clinical trials. In non-small-cell lung cancer and metastatic breast cancer, custirsen demonstrated potential in reducing CLU serum levels and improving patient survival. However, further larger phase III trials are needed to validate its benefit. In metastatic castration-resistant prostate cancer, custirsen did not show significant improvement in overall survival, and is not included in the standard treatment.

The new drug AB-16B5, a humanized monoclonal antibody targeting sCLU, underwent a phase I clinical study between 2015 and 2017 [https://clinicaltrials.gov/study/NCT02412462?intr=clusterin&page=2&rank=11 accessed on 10 August 2023]. The study involved patients with confirmed advanced solid malignancies that had not responded to prior therapies. The main objective was to assess the safety, pharmacokinetics and pharmacodynamics of AB-16B5. However, the results of this trial are yet to be published.

Currently, a phase II clinical study of AB-16B5 is ongoing and is scheduled to continue until the end of 2023 [https://clinicaltrials.gov/study/NCT04364620?intr=clusterin&page=1&rank=10 accessed on 10 August 2023]. In this phase II study, the drug AB-16B5 will be administered in combination with docetaxel to patients diganosed with metastatic non-small-cell lung cancer who have not experienced positive outcomes from traditional treatments. The aim is to evaluate the effectiveness and safety of this humanized monoclonal antibody. As of the current time, the results of the study have not been published, and we anticipate further investigation to potentially unveil new findings and insights.

## 6. Clusterin in the Eye: Relationship with Eye Disorders

The first publications about CLU expression in the eye, by the Jones group, appeared in 1992/1993 and reported an elevated CLU expression level during human retinis pigmentosa degenerative disorders [101,215]. Since then, numerous studies have been published on CLU expression in both the anterior and posterior segments of the eye, including the cornea, lens, ciliary body, retina, aqueous and vitrous [216,217,218,219] (Figure 6). Expression of CLU in various eye structures was subsequently described in developmental studies in mice, and confirmed CLU expression in retina (in the retinal pigment epithelium and endothelial cells). These studies also demonstrated different *CLU* mRNA patterns of expression during retinal development [164,165,220]. Indeed, in fetal mice, *CLU* mRNA is present in the retina (retinal pigment epithelium), lens and cornea [221]. Furthermore, in the adult mouse and human eye, *CLU* mRNA is present in the retina (inner nuclear and ganglion cell layers, and in the retinal pigment epithelium) and ciliary body [221]. Human *CLU* mRNA is also translated and CLU is secreted in aqueous and vitreous fluids of the eye. This suggests a local production of CLU from ganglion and inner nuclear layer cells to vitreous and from ciliary body cells to aqueous humor [221]. In primates, CLU expression was observed in lens, cornea, limbus, sclera, orbital muscle, ciliary body, retina, RPE/choroid and RPE cells in culture [217]. An analysis of the eye tissues in the National Eye Institute (NEI) bank libraries of the National Institutes of Health (NIH) (http://neibank.nei.nih.gov/index.shtml accessed on 1 August 2023) showed that the iris was the tissue where the *CLU* gene was the most abundantly expressed. CLU also appears important in lens, cornea and retina where it ranked 15th, 27th and 29th, respectively, among the most highly expressed genes [222].

CLU dysregulation has been shown to be beneficial in specific situations and detrimental in others. In the anterior segment, CLU has been associated with various conditions including ocular surface disorders, pseudoexfoliative glaucoma and corneal dystrophies [223,224,225,226]. CLU was also found to be a potential therapeutic option in wound healing. In the posterior segment, CLU was identified in age-related macular degeneration (AMD) as well as diabetic retinopathy, retinitis pigmentosa and Von Hippel–Lindau syndrome (Table 1).

### 6.1. The Anterior Segment of the Eye

#### 6.1.1. Dry Eye Disease (DED)

DED is one of the most common complaints when consulting an ophthalmologist, its prevalence ranging from 5–34% depending on the country, thus affecting hundreds of millions worldwide [247,248,249,250,251,252,253,254,255,256]. It is a multifactorial disease characterized by the loss of tear film homeostasis caused by its instability and hyperosmolarity, as well as inflammation and damage to the ocular surface and neurosensory abnormalities leading to symptoms like foreign body sensation or burning sensation [257]. Interestingly, low levels of CLU expression have been observed in the tear film of human DED patients and in Sjogren’s syndrome mice [127,232]. The interference of the epithelial turnover cycle results in the loss of apical cells and barrier disruption. Etiologies are numerous and include rosacea, blepharitis, Meibomian gland dysfunction, thermal and chemical burns, chronic use of glaucoma drops as well as autoimmune diseases such as Sjögren syndrome, graft vs. host disease and mucous membrane pemphigoid [258,259,260,261]. In Sjögren syndrome, epigenetic markers are altered and closely linked to progression of the disease [262]. In particular, histone demethylation (H3K4me2) increases PAX6 expression, leading to increased CLU expression in lacrimal glands and reversing disorders by improved tear film secretion [127]. Matrix metalloproteinase (MMP) activity, especially that of MMP9, has been linked with DED in several studies [263,264,265,266]; interestingly, CLU was found to be an inhibitor of MMP9, regulating its activity and therefore posing as a potential therapeutic target for DED [232,267]. Furthermore, a recent study established a correlation between low CLU concentration in tears and positive Schirmer strips test results (Schirmer strips being commonly used in research to assess tear production and therefore DED) [233]. Based on mice models, CLU was also hypothesized to bind selectively to damaged corneal surface and act as a wound healing mediator by sealing deficits in the epithelial barrier, preven- ting proteolysis due to MMP activation and reducing damage related to apoptosis and protein denaturation [234,268]. CLU was also found to promote corneal epithelial cell proliferation indirectly via upregulation of HGF (hepatocyte growth factor) [229]. CLU could possibly have a dampening effect on the immune response to disruption of the ocular surface as well as binding to damaged cells [232,234,269]. Based on these findings, CLU could offer therapeutic avenues in DED. Among other treatment alternatives for DED, serum-derived autologous eye drops have been developed based on the principle that tears contain large amounts of anti-inflammatory and pro-healing proteins which are also found in blood. These drops are effective in severe DED and are often the last therapeutic resort in late-stage disease. Their efficacy potentially derives from their concentration in CLU, thrice what is usually found in regular tears, illustrating the possibilities associated with CLU [233,270,271,272,273,274,275].

#### 6.1.2. Corneal Dystrophies

The corneal dystrophies are a group of mostly progressive and inherited diseases characterized by accumulation of extracellular deposits or intracellular cysts in different layers of the cornea, often impeding visual acuity and sometimes causing recurrent corneal erosions. CLU has been identified in the TGFBIp amyloid deposits of lattice corneal dystrophy type 1 [228] and gelatinous drop-like dystrophy [229], as well as in non-amyloid deposits in granular corneal dystrophy type 2 [228]. In recent years, CLU was found to be significantly elevated in Fuch’s endothelial corneal dystrophy (FECD). This condition is the most common cause of endogenous endothelial dysfunction resulting in loss of vision [230]. It is characterized by the deposition of extracellular collagenous material on the posterior surface of Descemet’s membrane as well as endothelial cell death and subsequent corneal oedema [276]. Although expression of CLU has been demonstrated in the normal human corneal endothelium [277], Jurkunas et al. showed that it was markedly increased in the endothelial cells from FECD patients for both nCLU and pre-secretory CLU (psCLU). However, sCLU levels were not significantly elevated. Overexpression of psCLU has been theorized to be a protective mechanism against apoptosis in stress-induced conditions [231] as higher concentrations of CLU often correlate with tissue insult [49,89,278]. In addition, higher levels of CLU are present in different diseases such as age-related macular degeneration and Alzheimer’s disease, both of which present with the accumulation of extracellular deposits followed by local cell death in a similar fashion as with FECD [279,280,281]. As oxidative stress plays a significant role in the pathophysiology of FECD [282,283], and CLU dysregulation also correlates with such stress, it is possible that these changes in CLU production are indirectly related to the mechanism involved in FECD [231].

#### 6.1.3. Stem Cell Culture and Transplantation on the Ocular Surface

Limbal stem cell deficiency (LSCD) results from damage to the epithelium and to the microenvironment of limbal epithelial stem cells, the most frequent causes being chemical or thermal burns, multiple surgeries involving the limbus, contact lens wear and inflammatory conditions of the ocular surface. LSCD leads to conjunctivalization of the cornea, thus resulting in opacification and impairment of vision [284,285]. Treatment options are limited since corneal transplantation yields poor results as epithelial turnover via the limbal stem cells is not effectively addressed by the usual surgical grafting of central cornea. Therefore, other alternatives were developed based on limbal stem cells transplantation. Epithelial cells extracted from the donor need to be cultured and expanded before they can successfully be grafted. In order to do so, they were initially seeded on mouse 3T3 fibroblasts feeder layers [286]. In 2011, Okada et al. investigated the effects of CLU on corneal epithelial cell growth by cultivating cells on 3T3 feeder layers transfected with a vector encoding full-length CLU. This transfection proved very beneficial to the colony-forming capacity of the cells via upregulation of HGF, a growth-promoting cytokine, by the feeder layers [229]. A year later, Mishima et al. described how CLU directly inhibited oxidative stress and oxidative-stress-induced damage in a side population of stem cells isolated from salivary and lacrimal glands of mice transplanted in a mouse model with medically-induced hypofunction of these glands via irradiation [287]. These recent findings illustrate how CLU might improve culture and transplantation of stem cells in the eye.

#### 6.1.4. Pseudoexfoliative Glaucoma

Pseudoexfoliation (PEX) syndrome is an age-related systemic disease in which ocular structures are mainly targeted. It manifests in the eye as the deposition of fibrillous white flaky material, described as pseudoexfoliative, originating from the lens, on the lens capsule, the iris, the ciliary body, the corneal endothelium and the pupillary margin [67]. Pseudoexfoliative glaucoma (PXG) happens in about 40% of affected people and is the most frequent cause of secondary open-angle glaucoma [288]. CLU has been identified as a component of PEX material in many studies [285,289,290] and low levels of *CLU* mRNA and CLU protein itself were found in the iris, lens and ciliary processes of patients with PEX syndrome [219,236]. Moreover, lower expression of CLU and PXE accumulation was shown to increase complement activation [291]. In most populations, variations of the gene encoding LOXL1, an enzyme responsible for cross-linking elastin fibers, have been strongly associated with PXG. In the Nordic population, two single nucleotide polymorphisms in LOXL1 provide a population attributable risk for PXG over 99%, although it does not apply to other populations in which the risk is different. In 2015, Fan et al. demonstrated that CLU variants might contribute moderately to the PXG risk, even though larger studies are required to confirm this finding [237].

### 6.2. The Posterior Segment of the Eye

#### 6.2.1. Age-Related Macular Degeneration (AMD)

AMD is the most common acquired retinal degeneration, mostly targeting people over 50 years old. Just over 200 million individuals suffer from AMD worldwide and more than 10 million in the United States alone [292,293]. It presents as a reduction in central vision by non-neovascular (called “dry” disease) and neovascular (called “wet” disease) changes in the macula. Up to 90% of affected individuals have dry AMD in which the manifestations include retinal atrophy and extracellular deposits between the retinal pigment epithelium and the Bruch’s membrane called “drusen” [294]. These drusen are the hallmark of AMD. They are composed of carbohydrates, zinc, lipids, complement factors and other cellular components, including CLU [239,240,241]. In 2005, Jackson et al. demonstrated a reduction in angiogenesis, the pathological state leading to neovascular AMD, when inhibiting CLU using antisense oligonucleotides [242]. Additionally, in 2021, Rinsky et al. found CLU levels to be elevated in the aqueous humor of patients with AMD when compared to controls, thus suggesting CLU may serve as a biomarker for the disease [235]. However, these results are somehow in contradiction with the demonstration that sCLU overexpression enhanced cell viability in human retina during retinal development and in AMD [164,165]. As some complement gene polymorphisms are linked to an increased risk of AMD [295,296,297,298] and alter autoregulation mechanisms of the complement system, it has been suggested that complement inhibitors could decrease the risk of AMD [299]. However, CLU’s role as a complement inhibitor has yet to be determined in AMD.

#### 6.2.2. Retinal Diseases

Diabetes mellitus (DM) is an endocrine disease in which abnormally high blood glucose can be measured. It is among the fastest growing and most common diseases worldwide. Projections for 2045 estimate the number of people with DM to be around 693 million adults globally [300]. Macrovascular and microvascular complications of hyperglycemia are a leading cause of morbidity and mortality [301]. Diabetic retinopathy (DR) is among the most frequent microvascular complications of DM as three out of four patients living with the disease for more than 15 years are affected [302]. Moreover, DR was the fifth leading cause of preventable blindness and moderate to severe visual impairment in people over 50 years of age in 2020 [303]. DR is also significantly associated with stroke, coronary artery disease and congestive heart failure [304]. Hyperglycemia induces vascular endothelial damage over time and impairs the function of the blood–retinal barrier (BRB) [305]. The BRB can be divided into an outer and an inner portion, the retinal pigment epithelium (RPE) and the retinal vascular endothelium, respectively, both of which possess tight junctions to maintain the barrier function. Its role is to strictly modulate the ins and outs of fluids and electrolytes in the extracellular space [306]. In DR, BRB dysfunction leads to retinal ischemia, neovascularization and diabetic macular oedema [305]. In 2007, Kim et al. demonstrated that supplementation of CLU to the culture of vascular endothelial cells could protect them from ischemia-induced damage and subsequent loss of tight junctions [220]. Three years later, the same team showed how CLU safely protected human RPE cells from oxidative-stress-induced apoptosis, a process dependent on signalization through the PI3K/Akt pathway [165]. Consistent with these findings, in vivo testing in a rat model with induced DR treated with intravitreal injections of CLU demonstrated that CLU had a BRB- and neural-protective effect via the prevention of tight junction loss and cell death by apoptosis. CLU thus proves to be a potential therapeutic agent in diseases with BRB dysfunction [243].

Based on CLU’s cytoprotective role in retinal degenerative diseases, the relationship between CLU and retinitis pigmentosa (RP) has been studied in recent years. RP is the most frequent retinal dystrophy, affecting 1 in 4000 individuals. Although all transmission patterns are described, most cases are sporadic. RP is characterized by retinal pigment deposits and the loss of rod photoreceptors, followed slowly by cone photoreceptors. Symptoms typically begin with nyctalopia and progress to cecity within decades [307]. In 2017, Vargas et al. studied the effect of CLU on rod photoreceptor survival in a rat autosomal dominant rhodopsin transgenic model of RP (S334ter-line-3). They found intravitreal CLU injections decrease greatly rod death at various postnatal times (30, 45, 60 and 75 days) when compared to saline-injected rats through activation of Akt and STAT3 as well as suppression of apoptosis-promoting BAX [244]. In a 2021 follow-up study, the American team showed that intravitreal CLU suppressed the upregulation of neuronal nitric oxide synthase (nNOS), a known cause of cone photoreceptor death, in RP retinas of a transgenic rat model (Rhodopsin S334ter-line-3) [245]. These findings therefore open therapeutic possibilities for CLU in RP.

The Von Hippel–Lindau (VHL) syndrome is a rare familial autosomal dominant multi-systemic disease characterized by numerous tumors affecting 14 different organs including the eye, the central nervous system, and the kidneys [308,309]. Retinal hemangioblastoma and hemangioma are the most common, and frequently the presenting, manifestations of VHL syndrome [309]. The disease is caused by mutations of the tumor suppressor gene VHL [310,311]. The VHL gene product (pVHL) is best known to target the transcription factor hypoxia-induced factor (HIF) for polyubiquitination and thus destruction [312,313,314]. In 2006, a study by Nakamura et al. showed the importance of CLU in VHL tumorigenesis and how CLU induction led to an HIF-independent pVHL function that could prove crucial for tumor suppression. A few years later, Zhou et al. found CLU to be markedly decreased in VHL-associated retinal hemangioblastoma and optic nerve hemangioblastoma tumor cells [246], which is consistent with previous findings in renal cell carcinoma and pheochromocytoma [315]. Thus, low levels of CLU expression might promote survival in hypoxic zones of VHL tumors [100]. Therefore, CLU could serve as a potential marker for pVHL function and reflection of VHL gene integrity in VHL-associated retinal hemangioblastoma and hemangioma [246].

#### 6.2.3. Uveal Melanoma (UM)

UM is the most common primary intraocular tumor in adults, representing 85% of all cancer of the eye and accounting for less than 5% of all melanoma cases [316,317]. Tumors appear when melanocytes from the uvea transform into melanoma cells. UM are mainly located in the choroid (85–90%), ciliary body (6%), or iris (4%) [317]. The incidence of UM has remained stable over the past 30 years (ranging from 0.2 to 9 cases per million depending on the geographical region) [317]. Despite excellent control of local disease with eye treatments, the prognosis remains poor due to metastatic progression which affects ~50% of patients within 6 to 12 months, with the liver involved in up to 90% of those cases [317]. In 2022, Pessuti et al. characterized the extracellular vesicles isolated from liquid biopsies of UM patients. They showed that the extracellular vesicle concentration was higher in analytes of UM patients compared to those of the control group. Samples from the aqueous humor, vitreous humor and plasma from UM patients were collected and showed an enrichment in extracellular vesicles of vitreous humor. Proteomic analysis was performed on them and the authors identified a panel of proteins that are mainly involved in protecting cells against apoptosis, controlling cell growth, promoting angiogenesis and inducing cell spreading, including CLU [318]. Thus, the increased concentration of extracellular vesicle composed of CLU could contribute to tumor progression, especially if it is sCLU. However, this study did not differentiate between the two CLU isoforms (nCLU vs. sCLU). Further studies will be required to evaluate this particular aspect. 

## 7. Clusterin and Wound Healing: The Tissue-Engineered Human Cornea as a Model

Studies that investigated the role of CLU in wound healing were most often conducted using cancer cells, including prostate cancer [319], hepatocellular carcinoma [320], laryngeal sqamous carcinoma [321], and both renal [223] and bladder cancer cells [322]. It has also been investigated in a few medical conditions including lung injury [323], renal inflammation and tissue repair [154,324], and in human cartilage repair [325]. A relationship between CLU levels in wound healing and immune response in Alzheimer’s disease has also been described [326]. However, to our knowledge, no such study prior to 2022 ever investigated the contribution of CLU to wound healing of any ocular structure, including the cornea. 

Due to its anatomical location (Figure 4), the cornea is the primary structure of the eye that undergoes changes when exposed to external aggressions, such as chemicals, heat, biological agents or physical trauma. In severe cases, conventional medical treatments involving anti-inflammatory and pro-healing substances may prove inadequate, necessitating corneal transplantation or even eye removal. In 2012, approximately 12.7 million individuals worldwide were awaiting corneal transplants. Unfortunately, only 185,000 corneal transplant surgeries were performed that same year, meeting the needs of merely 1 in 70 patients [230]. Consequently, the development of an entirely tissue-engineered human cornea (hTECs) assumes significant importance as it can serve as a model for studying wound healing mechanisms. This approach may offer alternative solutions to address the scarcity of suitable donor corneas. Over the past two decades, we have achieved success in producing a human tissue-engineered corneal substitute (hTEC) devoid of synthetic materials. Our hTEC exhibits a well-developed stratified epithelium composed of 5–7 layers of untransformed human corneal epithelial cells (hCECs) and a stroma. Due to its resemblance to the native cornea, our hTEC represents an exceptional model for in-depth investigation of cellular and molecular mechanisms involved in corneal wound healing.

In a study we recently published, the *CLU* gene was identified as one of the most significantly downregulated genes during the wound healing process of hTECs [115]. Considering the critical role of the extracellular matrix (ECM) in corneal repair and the established links between CLU and ECM components [146,238,327], these findings are not surprising. The involvement of CLU in cell adhesion, migration and proliferation is widely acknowledged, although the molecular mechanisms underlying its regulation remain unclear. Notably, CLU has a negative regulatory effect on fibronectin and type I collagen expression [238]. Therefore, the downregulation of CLU following hTEC injury aligns with the well-known increase in fibronectin levels that occurs early in the healing process [328,329]. The expression of matrix metalloproteinases (MMPs) is also profoundly affected by changes in ECM composition during corneal wound healing [330]. Our gene-profiling analyses revealed that MMP-9 and MMP-10 are among the 54 genes showing the most significant differential expression, with a robust increase in their expression under the wounded condition [115]. Interestingly, CLU has been reported to interact with MMP-9, inhibiting MMP-9-mediated breakdown of tight junctions between human epithelial cells [146]. Therefore, the observed decrease in clusterin expression in the wounded area corresponds to the increased expression of MMP-9 and MMP-10, which is consistent with the necessary ECM remodeling and cell migration during corneal regeneration. On the other hand, collagen promotes *CLU* gene expression [139]. Again, the decrease in collagen expression typically observed early in corneal wound healing [329] aligns with the reduced expression of CLU observed in our study, suggesting a potential regulatory feedback loop between ECM components and CLU expression.

In our hTECs, we demonstrated that both the transcription factors Sp1 and AP1 contribute to *CLU* gene transcription, their corresponding DNA target sites overlapping with one another within the *CLU* basal promoter [115]. However, Sp1 has a clearly greater affinity than AP-1 for binding its *CLU* target site, which is consistent with the coordinated decrease in the expression of both Sp1 and CLU that we observed during hTEC wound healing. In addition to the presence of the basal promoter, two, more distal, negative regulatory regions (silencers) were identified in the 5′ regulatory region of the human *CLU* gene. Several DNA target sites within the distal silencer (−1424/−2000) for TFs that have been documented as transcriptional repressors, such as AREB6, ARP-1, Bach1 and Evi1 [91,92,93,94,95,96], have been identified using tools for TF search. We believe it is plausible that the repressive function of these TFs, upon binding to their target sites in the *CLU* silencer, might partly contribute to the observed differential expression of the *CLU* gene in our hTEC model. However, further investigations are needed to confirm the binding of these negative TFs to the *CLU* distal silencer and establish their actual involvement in the repression of *CLU* gene expression during hTEC wound healing.

In summary, our work highlighted the significance of the *CLU* gene in corneal wound healing and provides insights into its regulation through interactions with ECM components and TFs. The findings contribute to our understanding of the cellular and molecular mechanisms involved in corneal repair and may have implications for developing novel therapeutic approaches for corneal injuries.

## 8. Concluding Remarks

The biological role of clusterin is undoutedly very complex, largely attributed to the presence of alternatively spliced isoforms, the distinct localization of their encoded protein products within and outside cells, as well as the specific expression and function of these isoforms in various cells and tissues. CLU appears to play diverse roles in the biological functions of individual cells, ranging from acting as a stress sensor to regulating tissue remodeling. In addition, no definitive mechanism has been proposed to account for the differential expression of these isoforms. Although the functions associated to CLU isoforms are sometimes complex to understand and most often tissue-specific, a common concept has emerged: nCLU and sCLU coexist in cells, and the precise regulation of their balance confers either pro- or anti-apoptotic properties. Both a decrease in sCLU and an increase in nCLU expression finally alter the nCLU/sCLU ratio and consequently contribute to promoting apoptosis in cells. CLU expression is ubiquitous but widely regulated in numerous pathologies. These include neurodegenerative diseases (Alzheimer’s and Parkinson’s diseases) and cancer. This explains why a significant proportion of the data on CLU regulation have been obtained in the context of cancer. These studies have highlighted the existence of a promising compound (OGX 011) which potentiates the effect of anti-cancer treatments. Currently, identifying the deregulation of CLU expression could serve as a biomarker, but more research is needed to understand the physiological role of CLU and its impact during wound healing, in order to open up therapeutic prospects, particularly in ophthalmology.

## Figures and Tables

**Figure 1 ijms-24-13182-f001:**
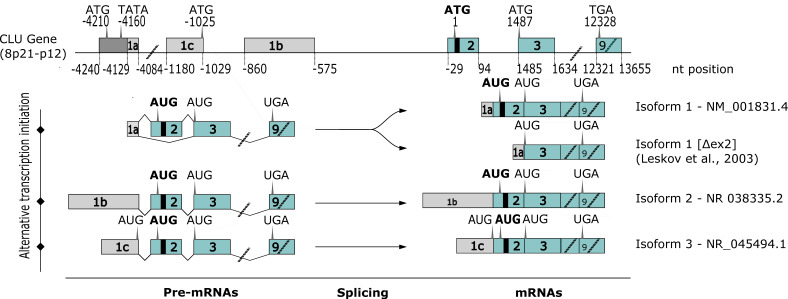
*CLU* gene and transcripts. Schematic representation of *CLU* gene organization on human chromosome 8 and of the three main mRNA variants. Green blocks represent remaining exons 2–9. Gray blocks represent unique exon 1 of the different *CLU* transcripts (isoform 1 NM_001831.4: exon 1a with 5′-extended sequence in dark gray, isoform 2 NR_038335.2: 1b and isoform 3 NR_045494.1: 1c). Transcriptional start sites are represented with ATG (or AUG in mRNA). Alternative splicing of variant 1 pre-mRNA generates an mRNA (isoform 1 [Δex2] [23]) that lacks exon 2 and the leader peptide sequence (black box). The position of the *sCLU* start codon (framed) is defined as nt = 1. Notice the additional in-frame AUG codons on exon 3 of all mRNAs and on exon 1c of variant 3. The canonical transcription start site of *CLU* mRNA isoform 1 is located at the begining and 23 nucleotides downstream of the TATA promoter element. The termination codon UGA is located in exon 9 at position 12,328 nt.

**Figure 2 ijms-24-13182-f002:**
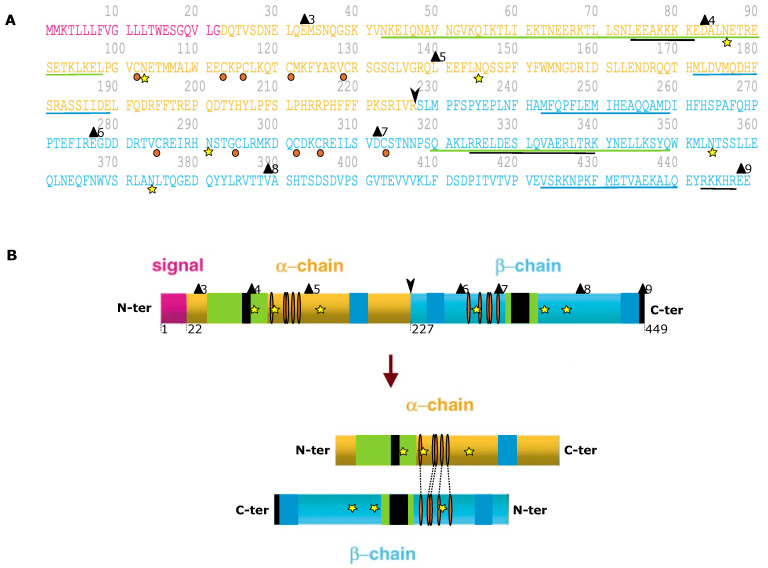
(**A**) Predicted translation product of the most abundant transcript (isoform 1), and (**B**) schematic representation of human psCLU protein precursor structure (top) and mature protein (bottom). The precursor polypeptide chain (top) is cleaved proteolytically to remove the 22-mer secretory signal peptide (pink) and subsequently cleaved (black arrow) between Arg 227 and Ser 228 to generate the α (yellow) and β (light blue) chains. These are assembled in anti-parallel orientation to give a heterodimeric molecule (bottom) in which the cysteine-rich centers (brown) are linked by five disulfide bonds (brown ellipses in panels (**A**,**B**)) and are flanked by two predicted coiled-coil α-helices (green) and three predicted amphipathic α-helices (dark blue). The six sites of N-linked glycosylation are indicated as yellow stars. nCLU contains 3 nuclear localization signals (NLS) (black) that relocate the nCLU protein from the cytoplasm to the nucleus. The predicted secreted protein is a 449-amino-acids sequence (adapted from ref. [41]).

**Figure 4 ijms-24-13182-f004:**
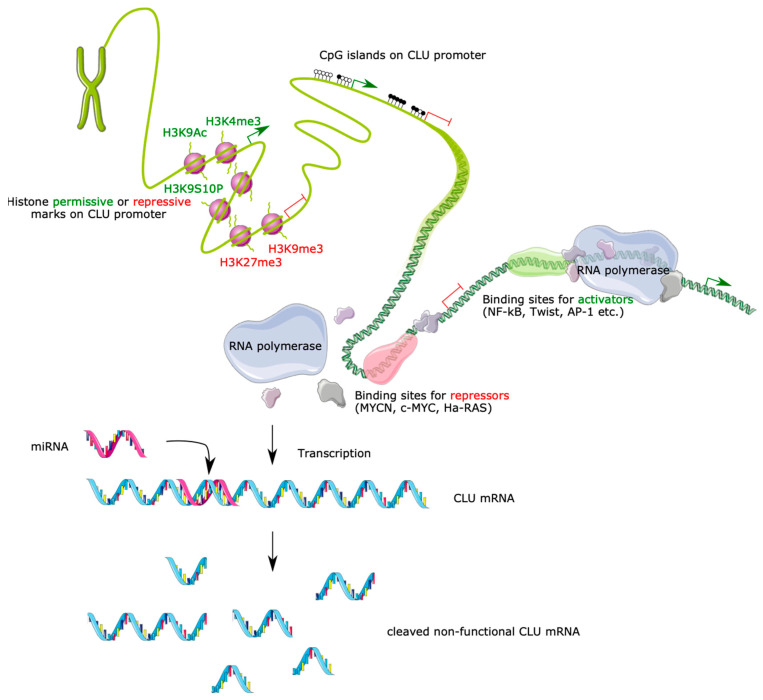
Main regulatory mechanisms of *CLU* expression. *CLU* gene can be regulated in different ways. DNA is coiled around histone proteins, allowing it to be efficiently compacted. The modulation of *CLU* gene expression is closely associated with specific histone modifications. Permissive histone marks, such as H3K9Ac, H3K9S10P and H3K4me3, as well as repressive marks, such as H3K27me3 and H3K9me3, present on the *CLU* promotor region regulate *CLU* transcription. The addition of methyl groups to the CpG island that alter transcriptional activity without any changes to the DNA sequence is also another *CLU* regulatory mechanism. Methylation of the *CLU* promoter region (black dots) is associated with weak CLU protein expression, while hypomethylation (white dots) allows *CLU* gene expression. Transcriptional control of the *CLU* gene transcription can also involve key transcription factors that bind to specific regulatory DNA sequences present on the *CLU* gene promoter to either repress (repressors; red bars) or induce (activators; green arrows) *CLU* gene transcription. In addition, microRNAs (miRNAs) are small single-stranded, non-coding RNAs that post-transcriptionally regulate mRNA expression by base-pairing with complementary sequences within *CLU* mRNA transcripts. As a result, these mRNA molecules are silenced through multiple mechanisms including cleavage of the mRNA strand, destabilization of the mRNA or a reduced translation efficiency of the mRNA into proteins.

**Figure 5 ijms-24-13182-f005:**
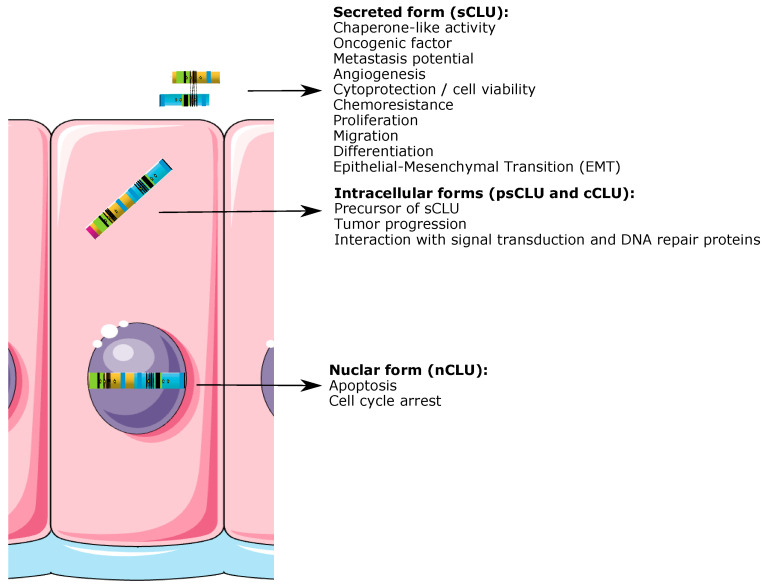
CLU isoforms and functions (see legend to Figure 2 for the color code of CLU proteins).

**Figure 6 ijms-24-13182-f006:**
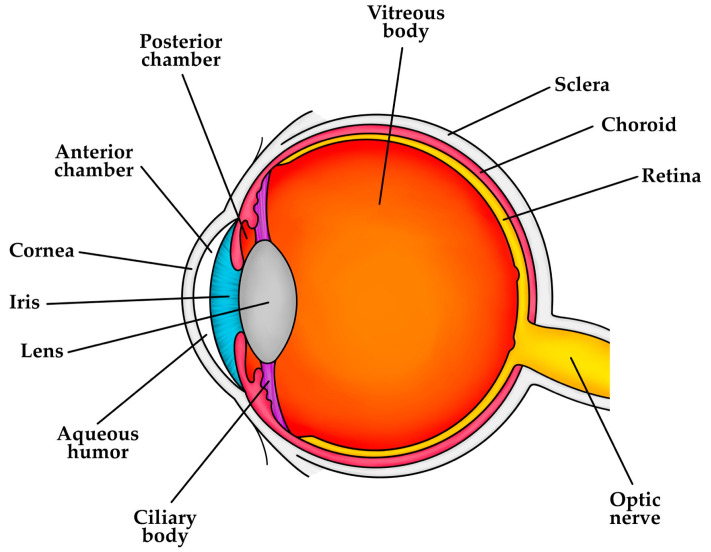
Anatomy and structure of the human eye.

**Table 1 ijms-24-13182-t001:** CLU expression and function in ocular pathologies.

Structure	CLU Expression	Pathology	Function
**Cornea**	Yes	LSCD	CLU increase colony forming efficiency of epithelial limbal stem cells [227]
		Dystrophies (lattice corneal dystrophy type 1, gelatinous drop-like dystrophy, non-amyloid deposits in granular corneal dystrophy type 2 and Fuch’s dystrophy)	CLU expression in amyloid deposits [228,229] and guttea [230] nCLU et psCLU surexpression in FECD cells [231] psCLU accumulation in cytosol potentially protect against apoptosis induce by oxidative-stress [231]
		DED	Low level of CLU in tear film leading to loss of apical cells and barrier disuruption [127,232] CLU improve tear film secretion [127] and seal deficits in the epithelial barrier [233,234]
**Anterior chamber (aquous humor)**	Yes	AMD	CLU enrichment [235]
**Iris**	Yes	PXG	Low expression of CLU [219,236] CLU variants increase PXG risk [237]
**Posterior chamber**	Yes		
**Ciliary body**	Yes	PXG	Low expression of CLU [219,236] CLU variants increase PXG risk [237]
**Vitreous body**	Yes	UM	Increased extracellular vesicle concentration composed of CLU [238]
**Lens**	Yes	PXG	Low expression of CLU [219,236] CLU variants increase PXG risk [237]
**Retina**	Yes	AMD	Drusen composed of CLU [239,240,241] CLU inhibition reduces neovascularisation in wet disease [242] sCLU enhanced cell viability [164,165]
		DM	CLU protect from ischemia-induced damage and loss of junctions [220] CLU protect RPE cells from oxidative stress-induces apoptosis [165] CLU participe to the BRB function via the prevention in tight-junction loss and cell death [243]
		RP	CLU ehnanced rod photoreceptors survival [244] CLU decrease cone photoreceptors death by suppressing nNOS expression [245]
		Hemangioblastoma	CLU decrease [246]
**Choroïd**	Yes		
**Sclera**	Yes		
**Optic nerve**	Yes	Hemangioblastoma	CLU decrease [246]

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
