# Peer review of "The Ins and Outs of Clusterin: Its Role in Cancer, Eye Diseases and Wound Healing"

_ijms, 2023, doi:10.3390/ijms241713182_

Round 1

Reviewer 1 Report

The review article with the title “The Ins and Outs of Clusterin: Its Role in Cancer, Eye Diseases and Wound Healing” is an extremely informative article that critically reviews current scientific knowledge about clusterin. Two major and two minor comments can be found below for the improvement of the manuscript.

·         I would suggest including 2 or 3 figures summarizing the role of clusterin. For example, one figure about its regulation, one about its regulatory potential, and one about its involvement in the eye. Otherwise, the information appears extensive and the reader can not properly summarize it. Furthermore, I believe that Figure 4 does not extremely correlate with the main topic of this review.

·         As it is an extended article, I would strongly recommend excluding explanations about well-known terms. For example, the first paragraphs in the “Modification of histones” section could be better summarized.

·         Figure 1 appears compressed.

·         A few minor language issues can be found.

Only a few minor typing issues can be found in the manuscript.

Author Response

Comment 1: I would suggest including 2 or 3 figures summarizing the role of clusterin. For example, one figure about its regulation, one about its regulatory potential, and one about its involvement in the eye. Otherwise, the information appears extensive and the reader can not properly summarize it. Furthermore, I believe that Figure 4 does not extremely correlate with the main topic of this review. 

Response (Figure 4 to 6): We thank the reviewer for this suggestion. We decided to keep Figure 6 (initially figure 4) intact as not all the readers are familiar with the human eye structures (both the aqueous humor and the vitreous body were however added as additional information). However, we added a table (Table 1) that describes the multiple involvements of CLU in the eye. We hope this will facilitate understanding of the corresponding text.

We have also added two new figures (Figures 4 and 5). The first (figure 4) summarizes the different modes of regulation of CLU expression described in our article. The second (figure 5), attempts to clarify the role of each CLU isoform according to its localization in the cell.

Comment 2: As it is an extended article, I would strongly recommend excluding explanations about well-known terms. For example, the first paragraphs in the “Modification of histones” section could be better summarized.

Response: As the aim of our article attempted to raise awareness of the importance and the complexity of CLU, we wrote a first part about generality (1.introduction, 2.gene and protein, 3.clusterin gene regulation). Then, we focussed on processes involved in wound healing (4. Clusterin in tissue remodeling, 5. Clusterin and cancer,) and eye diseases (6.Clusterin in the eye: relationship with eye disorders), our specific field of expertise. As the first part is very general, we are aware that some of our readers would not be specialists in this field, so we decided to use general paragraphs at the beginning of each section to help everyone understand.

Comment 3: Figure 1 appears compressed.

Response: The previous figure has been replaced by an uncompressed version. We also provided a separate pdf of the figure together with this submission.

Comment 4: A few minor language issues can be found.

Response: The entire manuscript has been carefully reviewed for minor spelling mistakes.

Reviewer 2 Report

Manuscript number: IJMS-2508946

Title: The Ins and Outs of Clusterin: Its Role in Cancer, Eye Diseases and Wound Healing

General comments

This review presents a detailed overview of clusterin, a glycoprotein involved in numerous biological processes. In addition to the genetic structure, the different isoforms are described and the regulatory mechanisms by micro RNAs. The authors show that clusterin is involved in various signaling pathways. The review describes the role of clusterin in cancer and various eye diseases. Overall, the review provides a broad overview, with numerous references for further information.

Author Response

Reviewer 2 did not require any modification to the text. We would like to thank this reviewer for his/her very constructive comments on our review article.

Reviewer 3 Report

Ch.Gross et al. submitted a review article on clusterin (CLU). This protein has numerous synonyms reflecting the wide array of functions of in health and diseases. It is therefore not always easy to focus on all aspects of CLU in a review of reasonable length. In this review emphasis is put on CLU and its role in tumorigeneis and cancer, and in part also of its role in eye diseases.

Comments

This is a very interesting review on the function and pathophysiology of clusterin. Because of the wide distribution in human organs and its potential role in numerous diseases it is obviously not possible to treat all aspects in detail. Nevertheless there are three pints that need considerations:

1.     The cited literature is somewhat old, and what one misses are references to more recent publications. In fact there are only -3 references from 2022 and 2023, and the reader might get the impression that this is merely a warm-up of an earlier review or a PhD thesis.

2.     Although it might be impossible to list all functions and pathological effects of CLU, one misses a chapter related to CLU and its role in brain function and diseases of the brain such as Alzheimer. One paragraph on this topic would be most welcome.

3.     One also wonders how much work had been devoted by the authors to clusterin as there are only 2 publications of them in the reference list related to CLU.  

Minor point: Fig. 1 in the PDF copy is black and white  - yet the authors refer to some colored boxes in the legend of this figure.

Author Response

Comment 1: The cited literature is somewhat old, and what one misses are references to more recent publications. In fact there are only -3 references from 2022 and 2023, and the reader might get the impression that this is merely a warm-up of an earlier review or a PhD thesis.

Response: In the initial review, there was 10 articles from 2020 and 11 from 2021. As mentioned by the reviewer 3, only few references were from 2022 and 2023 (respectively 1 and 3). As requested, we added new references (2 published in 2020, 1 in 2021, 6 in 2022 and 2 in 2023) and discussed them accordingly in the appropriate sections of the paper (lines 613 – 626; lines 732 – 741; lines 956 – 969; lines 1041 - 1052; lines 1062 – 1076; lines 1246 – 1263; lines 1278 – 1280; lines 1284 – 1288; lines 1460 - 1479). Furthermore, during our research into the progress of CLU research in 2022 and 2023, we discovered a new product currently in clinical trials phase I that regulates CLU expression. We therefore decided to add a paragraph on this subject (lines 1216 – 1230).

Comment 2: Although it might be impossible to list all functions and pathological effects of CLU, one misses a chapter related to CLU and its role in brain function and diseases of the brain such as Alzheimer. One paragraph on this topic would be most welcome.

Response: This is an excellent suggestion. However, we felt such addition would not fit with the actual subject of this review, which focusses on CLU expression in wound healing and eye diseases. In this review, the theme of cancer is addressed for its common features with wound closure. That's why only cancer is explored in our manuscript. Other pathologies involving or modulating CLU expression are only briefly touched on to explain their importance in relation to the migratory, proliferative, etc. components of healing. Furthermore, there are already recent reviews on the subject of neurodegenerative diseases that pertinently explain the role and function of CLU : Yuste-Checa et al, 2022 (Alzheimer), Milinkevicuite G et al, 2023 (Alzheimer), Chaplot et al, 2020 (Neurodegenerative diseases).

Comment 3: One also wonders how much work had been devoted by the authors to clusterin as there are only 2 publications of them in the reference list related to CLU.

Response: We began our CLU study in 2020 when Christelle Gross started her postdoc in the lab. Based on genetic profiling data performed in our lab, we identified the CLU gene as being differentially expressed following eye injury and decided to investigate further what could account for this deregulated CLU expression. To get started in this field and because of Covid-19, Christelle did an exhaustive review of the literature as soon as she arrived. In 2022, after her first experimental publication on the CLU gene in the cornea, we decided to pass on to the scientific community the knowledge we gained from her research over the years on this fascinating gene.

Comment 4: Minor point: Fig. 1 in the PDF copy is black and white- yet the authors refer to some colored boxes in the legend of this figure.

Response: Thank you for the comment. In our computer, the Figure appears in color. We therefore have no clue as to why it shows as a black and white figure in your version of the manuscript, maybe during the MDPI manuscript assembling process? To avoid any further disagreement, all figures were deposited on the IJMS web site also as separate PDF documents.

Round 2

Reviewer 3 Report

  I evaluated the revised version again in view of the comments you sent.    Actually I have no further concerns and ask the Editor to accept the paper as it stands.

I cinsider the English mostly OK